# Decreased liver B vitamin-related enzymes as a metabolic hallmark of cancer cachexia

Yasushi Kojima [1] ✉, Emi Mishiro-Sato [1], Teruaki Fujishita [1], Kiyotoshi Satoh[2], Rie Kajino-Sakamoto [1], Isao Oze [3], Kazuki Nozawa[4], Yukiya Narita[4], Takatsugu Ogata[4], Keitaro Matsuo [3], Kei Muro[4], Makoto Mark Taketo[5], Tomoyoshi Soga [2] & Masahiro Aoki [1,6] ✉

Cancer cachexia is a complex metabolic disorder accounting for ~20% of cancer-related deaths, yet its metabolic landscape remains unexplored. Here, we report a decrease in B vitamin-related liver enzymes as a hallmark of systemic metabolic changes occurring in cancer cachexia. Metabolomics of multiple mouse models highlights cachexia-associated reductions of niacin, vitamin B6, and a glycine-related subset of one-carbon (C1) metabolites in the liver. Integration of proteomics and metabolomics reveals that liver enzymes related to niacin, vitamin B6, and glycine-related C1 enzymes dependent on B vitamins decrease linearly with their associated metabolites, likely reflecting stoichiometric cofactor-enzyme interactions. The decrease of B vitamin-related enzymes is also found to depend on protein abundance and cofactor subtype. These metabolic/proteomic changes and decreased protein mal-onylation, another cachexia feature identified by protein post-translational modification analysis, are reflected in blood samples from mouse models and gastric cancer patients with cachexia, underscoring the clinical relevance of our findings.

Cancer cachexia is a multifactorial syndrome characterized by an ongoing, involuntary loss of body weight and skeletal muscle mass[1]. It reportedly affects 50–80% of advanced cancer patients[2,3] and is believed to be responsible for 1–2 million annual deaths worldwide annually[4]. However, developing early diagnostic methods and effective therapeutic interventions for cancer cachexia has been challenging, likely due to an incomplete understanding of its pathophysiology.

Cancer cachexia has been conceptualized as a complex metabolic disorder related to systemic inflammation[5,6]. Although several proinflammatory cytokines and myokines have been implicated in cachexia, clinical trials targeting them have not demonstrated clinical benefits so far[7,8]. Nutritional interventions appear to be ineffective in controlling cancer cachexia[9]. Previous studies have demonstrated the roles of skeletal muscle and adipose tissue where atrophy is observed in cancer cachexia[3,6,8], but the key metabolic features of cancer cachexia remain unidentified. This situation contrasts with diabetes mellitus, in which identification of the disease as hyperglycemia caused by defects in insulin secretion and/or action has facilitated development of treatments[10]. Here, we explore the metabolic landscape of cancer cachexia and specify its metabolic hallmarks, based on an integrative analysis of multiple-omics data from several mouse models of cancer and gastric cancer patients.

[1]Division of Pathophysiology, Aichi Cancer Center Research Institute, 1-1 Kanokoden, Chikusa-ku, Nagoya, Aichi 464-8681, Japan. [2]Institute for Advanced Biosciences, Keio University, 246-2 Mizukami, Kakuganji, Tsuruoka, Yamagata 997-0052, Japan. [3]Division of Cancer Epidemiology and Prevention, Aichi Cancer Center Research Institute, 1-1 Kanokoden, Chikusa-ku, Nagoya, Aichi 464-8681, Japan. [4]Department of Clinical Oncology, Aichi Cancer Center Hospital, 1-1 Kanokoden, Chikusa-ku, Nagoya, Aichi 464-8681, Japan. [5]Colon Cancer Project, Kyoto University Hospital-iACT, Kyoto University, Yoshida-Konoe-cho, Sakyo-ku, Kyoto 606-8501, Japan. [6]Department of Cancer Physiology, Nagoya University Graduate School of Medicine, 65 Tsurumai-cho, Showa-ku, Nagoya, Aichi 466-8550, Japan. ✉e-mail: ykojima@aichi-cc.jp; msaoki@aichi-cc.jp

## Results

### Metabolomic analysis reveals alterations of metabolites related to NAD and one-carbon (C1) metabolism in severe cancer cachexia models

We employed multiple mouse models for this study to identify metabolic features of cancer cachexia while avoiding results limited to a particular model or cancer type (Table S1). MEWO and SEKI are human melanoma xenograft models[11]. CRCA (cis-Apc[+/Δ716]/Smad4[+/-] mice) and LNCA (K-ras[LSL-G12D/+]/p53[lox/lox] mice) are genetically engineered mouse models for colorectal and non-small cell lung cancer, respectively[12–14]. A starvation experiment (STRV) was included to produce body compositional and metabolic changes typical of simple fasting[15]. Analyses of survival, body weight, food intake, body composition, muscle atrophy markers, and plasma proteins indicated that SEKI, LNCA, and CRCA undergo body weight loss with muscle wasting and reduced food intake, phenotypes characteristic of cancer cachexia (Fig. S1a-e). Several clinical studies have reported that the increase of C-reactive protein (CRP), a liver-derived positive acute phase protein (APP), and a decrease of albumin, a negative APP, correlates with severity of cachexia and negatively with prognoses of cancer patients[16,17]. SEKI and CRCA blood samples contain larger quantities of a positive APP, serum amyloid A (SAA), compared with those of other models, whereas serum albumin is reduced in CRCA alone (Fig. S1e). Together with degrees of muscle loss and upregulation of muscle atrophy markers (Fig. S1c, d), these results suggest an increasing gradient of the severity of cachexia-related symptoms in LNCA, SEKI, and CRCA (Fig. 1a, b). We observed no substantial sex differences in survival time or body compositional changes (Fig. S2a, b).

Next, we performed capillary electrophoresis-time-of-flight mass spectrometry-based (CE-MS) metabolome profiling of skeletal muscle and liver of five models and their controls[18,19]. Volcano plots indicated the highest differential distribution of metabolites in CRCA, followed by SEKI, among the four cancer mouse models (Fig. 1c). Degrees of metabolic fluctuation also supported an increasing gradient of cachexia severity from MEWO to CRCA. Principal component analysis (PCA) of CE-MS data, body compositional data, and the use of a linear support vector machine (SVM) algorithm marked a potential boundary between the severe cachexia models (SEKI and CRCA) and other cancer models (MEWO and LNCA) (Fig. 1d).

To specify metabolic features of cancer cachexia, we next employed the Random Forests learning method and selected metabolites that showed similar changes between severe cachexia models, SEKI and CRCA (Fig. 1e)[20]. The list was enriched in metabolites related to nicotinamide adenine dinucleotide (NAD) and one-carbon (C1) metabolism. NAD, a water-soluble B vitamin metabolite, is essential in numerous redox reactions as a coenzyme and signal transduction as a substrate[21,22]. C1 metabolism has been conceptualized as a metabolic system that exchanges C1 units, such as methyl groups[23]. Notably, the liver is the central hub maintaining systemic homeostasis of both NAD and C1 pathways[21,24].

### Niacin and vitamin B6 decrease in the liver metabolome of SEKI and CRC

NAD synthesis in the liver is responsible for supplying peripheral tissues with nicotinamide and maintaining systemic NAD homeostasis[25]. NAD was decreased in muscle and liver of our four cancer-bearing models (Fig. 2a; Table S2)[13,26–29]. Notably, we found a downward trend in NAD from LNCA to CRCA (Fig. 2a; Table S3). In addition, liver metabolome data showed a decrease in NADP, an NAD derivative, and nicotinamide (NAM), the NAD precursor, in severe cachexia models, SEKI and CRCA (Fig. 2a; Table S2). These data suggest that this alteration of NAD metabolism is present in muscle and liver of cancer-bearing and is associated with cacheixa severity.

NAD-related metabolites belong to a B vitamin subgroup, niacin (NIA), also known as vitamin B3[22]. When we summed NAD-related metabolites and regarded the aggregate value as NIA, we found a significant downward trend in hepatic NIA levels from LNCA to CRCA (Fig. 2b; Table S4). A similar downward trend was found in vitamin B6 (B6) levels in liver (Fig. 2c; Table S4), where B6-related metabolites are primarily metabolized[30]. To address the possible effect of B vitamin supplements on the survival of severe cachexia models, we administered a high-dose B vitamin cocktail, including NIA and B6, to SEKI mice[22]. However, the cocktail showed no effects on their survival (Fig. S3a).

### Glycine and betaine, highly abundant C1 metabolites, are reduced, whereas less abundant S-adenosyl methionine is maintained in SEKI and CRCA liver metabolomes

We next investigated C1 metabolism, the other metabolic pathway altered in mice with cachexia. Among major C1 metabolites in the liver (Fig. 2c), Gly and betaine (tri-methyl glycine) are more abundant than others, such as S-adenosyl methionine (SAM), methionine, and folate (Fig. 2d). Our metabolomic data showed a decrease in hepatic betaine and Gly in severe cachexia models, SEKI, and CRCA (Fig. 2e; Table S2). Consistent with a previous report using a different mouse model of cancer cachexia, daily administration of high-dose Gly to SEKI mice resulted in significant prolongation of their survival (Fig. S3b)[31]. However, it caused no significant changes in body weight, food intake, or tumor growth, and the underlying mechanism remains unclear (Fig. S3c).

Next, we examined another key metabolite of C1 metabolism, SAM. SAM donates a methyl group to a target molecule and is converted to S-adenosyl homocysteine (SAH). Liver is responsible for ~85% of SAM-dependent methylation reactions required throughout the body[24]. PCA of proteomic data for SAM-related enzymes isolated the liver from a cluster of data points for 27 other tissues, supporting the unique role of the liver in SAM metabolism (Fig. S3d)[24,32]. While SAM levels tended to decrease in muscle along with increasing severity of muscle loss and expression of muscle atrophy markers (Fig. S1c, d), they tended to be maintained in livers of SEKI and CRCA mice (Fig. 2f; Table S2).

Our integrated analysis of metabolomic datasets from three models (LNCA, SEKI, and CRCA), highlighted correlations between cachexia severity and specific metabolite changes (NAD, NIA, B6, betaine, glycine, and SAM). To further confirm these correlations between cachexia severity and metabolites, we analyzed a CRCA dataset, which exhibited heterogeneity of cachexia severity, because it included mice under 10 weeks of age that did not develop severe cachexia. Scatterplot analysis showed linear relationships between an indicator of cancer severity, muscle weight change, and changes in metabolites (NAD, NIA, B6, betaine, and glycine levels, and SAM/SAH ratio) in CRCA, reconfirming that these metabolic changes are correlated with cachexia severity (Fig. S3e). In summary, our metabolomic data showed cancer cachexia-related metabolic changes in niacin, vitamin B6, and C1-related metabolites (Fig. 2g).

### Hepatic C1 metabolism is closely linked to two B vitamin subgroups (niacin and vitamin B6)

Our metabolomic data showed concomitant metabolic changes of two B vitamins (niacin and B6) and C1 in cachectic liver. We then explored a metabolic link between the two pathways using anti-tumor compounds as simple metabolic disturbing chemicals. Treatment of wild-type C57BL/6N mice with FK866, a highly selective inhibitor of nicotinamide phosphoribosyltransferase (NAMPT) that blocks NAD production[33], not only reduced muscle and liver NAD and NIA levels, but also affected C1 metabolism, causing a decrease in the liver Gly level and an increase in the liver SAM/SAH ratio (Fig. S4a; Table S2). Moreover, FK866 treatment significantly reduced vitamin B6, suggesting a close metabolic connection between NIA and vitamin B6 (Fig. S4a; Table S2). 5-fluorouracil (5FU) is a potent inhibitor of thymidylate

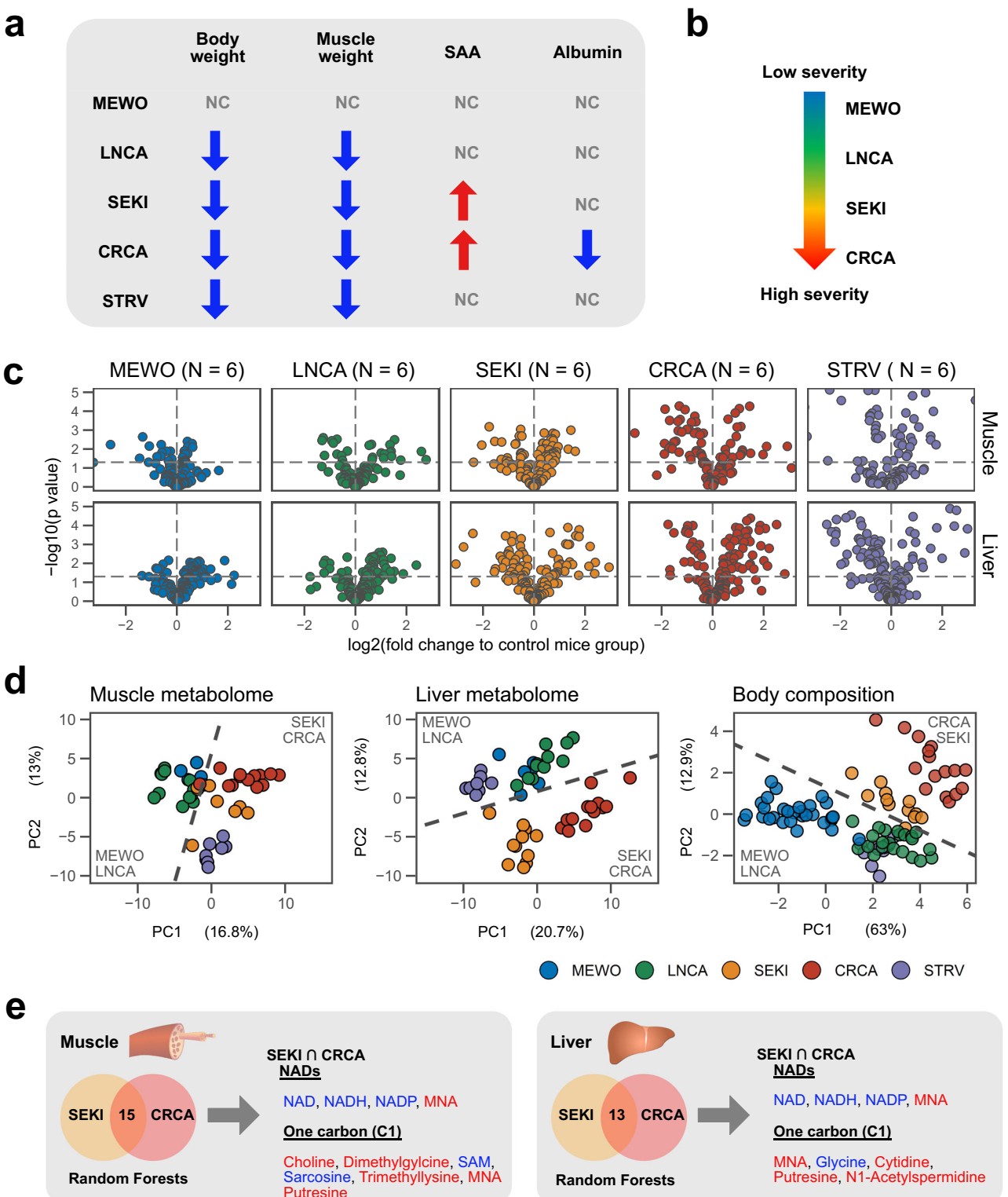

synthase, which is involved in C1 metabolism[34]. 5FU treatment decreased muscle and liver Gly levels, whereas the SAM/SAH ratio was decreased in muscle, but increased in liver. 5FU treatment also reduced muscle NIA levels of wild-type C57BL/6N mice (Fig. S4a; Table S2). Notably, these two drugs, FK866 and 5FU, increased the SAM/SAH ratio in liver, but not in muscle (Fig. S4a; Table S2). In addition, our integrated pooled mouse liver metabolomic data showed that SAM and its derivatives, 1-methyl nicotinamide (MNA) and putrescine, were negatively related to NIA and B6 (Fig. S4b, c; Table S2,

S3)[35,36]. Collectively, these results imply that hepatic C1 metabolism is inextricably linked to these B vitamin subgroups (niacin and vitamin B6).

**Proteomic analysis reveals more drastic hepatic proteomic changes in SEKI and CRCA**

To capture proteomic features associated with cancer cachexia, we conducted a quantitative proteomic analysis using liquid chromatography-mass spectrometry (LC-MS). Linear SVM algorithms

**Fig. 1 | Metabolomic analysis reveals alterations of metabolites related to NAD and one-carbon (C1) metabolites in severe cancer cachexia models. a** Summary of cancer cachexia-related phenotypes. SAA, serum amyloid A. Blue arrows, decrease; red arrows, increase; NC, no significant change. Details of mouse models used in this study are described in Table S1 and Fig. S1. **b** Schematic of our four mouse cancer models showing gradation from MEWO to CRCA in severity of cachexia. **c** Volcano plots of muscle and liver metabolomic data. Horizontal dashed lines, $P = 0.05$ (empirical Bayes test). MEWO, $N = 6$ mice; LNCA, $N = 6$ mice; SEKI, $N = 6$ mice; CRCA, $N = 6$ mice; STRV, $N = 6$ mice. **d** PCA plots of muscle metabolomic, liver metabolomic, and body compositional data (changes in body weights and weights of gastrocnemius muscles, anterior tibialis muscles, soleus muscles, liver, subcutaneous and visceral white adipose tissue, brown adipose tissue, kidney,

and spleen). Contribution ratios (%) of the first (PC1) and second principal components (PC2) are indicated in the respective axis titles. Dashed lines, hyperplanes produced by linear SVM between severe cachexia models (SEKI and CRCA) and other cancer models (MEWO and LNCA). Muscle metabolome; MEWO, $N = 6$ mice; LNCA, $N = 11$ mice; SEKI, $N = 8$ mice; CRCA, $N = 13$ mice; STRV, $N = 8$ mice. Liver metabolome; MEWO, $N = 6$ mice; LNCA, $N = 11$ mice; SEKI, $N = 11$ mice; CRCA, $N = 13$ mice; STRV, $N = 8$ mice. Body composition; MEWO, $N = 30$ mice; LNCA, $N = 22$ mice; SEKI, $N = 12$ mice; CRCA, $N = 16$ mice; STRV, $N = 12$ mice. **e** Overlap of key metabolites selected by Random Forests (mean decreasing accuracy (MDA) > 1.5). Venn diagram showing numbers of metabolites. Red, upregulated; blue, downregulated. NAD nicotinamide adenine dinucleotide; MNA, 1-methylnicotinamide, SAM S-adenosyl methionine. Source data are provided as a Source Data file.

separated severe cachexia models (SEKI and CRCA) from other cancer models (MEWO and LNCA) in the muscle and liver PCA plots (Fig. S5a). Enrichment analyses of PC loadings suggested changes in mitochondrial proteins in muscle and liver of SEKI and CRCA mice (Fig. S5b), and volcano plots showed escalating fluctuations of mitochondrial proteins from MEWO to CRCA (Fig. S5c). Next, to address whether metabolomic features of cancer cachexia are associated with proteomic changes, we selected NIA-, B6, and C1-related enzymes, based on annotation information of the UniProt database (Table S5)[37], for further analysis. Consistent with the metabolomic data, concentrations of NIA-, B6-, and C1-related enzymes showed more significant fluctuations in liver than in muscle (Fig. S5d), and these hepatic fluctuations were much higher in SEKI and CRCA than in MEWO and LNCA.

To further verify that cachexia-associated proteomic changes were more intense in liver than in muscle, we generated one thousand adjusted liver proteome datasets by computer-assisted extraction of 1057 proteins from the original datasets so that each dataset contains the same numbers of proteins belonging to the mutually exclusive compartments as the muscle dataset, that is, 96 NIA + B6 + C1, NIA-, B6-, and C1-related enzyme proteins, 292 other enzymes, and 669 non-enzymatic proteins (Fig. S5e). To compare the muscle and adjusted liver datasets, we performed PCA. Linear SVM algorithms consistently separated severe cachexia models (SEKI and CRCA) from other cancer models (MEWO and LNCA) with more extended maximum margins in the liver PCA plot panels than in those of muscle (Fig. S5f), confirming more drastic proteomic changes in liver than in muscle. These PCA plots indicate that the liver, rather than muscle, is the site of major metabolic disturbances characteristic of severe cachexia.

### Cachectic liver shows structural and proteomic changes, accompanied by overproduction of acute-phase proteins

Since enrichment analysis of liver proteome data suggested intracellular organelle changes (Fig. S5b), we examined the ultrastructure of cachectic liver. Electron microscopic images exhibited severe structural and compositional changes of organelles in SEKI and CRCA hepatocytes (Fig. 3a–c). Specifically, they showed swelling of mitochondria, reduced lipid droplets, and expansion of endoplasmic reticula (ER) with ribosomes, compared with control, MEWO, and LNCA hepatocytes[38,39].

Organelle-based volcano plots indicated upregulation of protein synthesis machinery, specifically, ribosome, ER, and Golgi apparatus, in SEKI and CRCA livers (Fig. 3d). Furthermore, we observed upregulation of immunity-related proteins, including acute-phase proteins (APPs), and downregulation of liver-specific enzymes, in SEKI and CRCA livers. We suspect that overproduction of APPs in cachectic liver may result in a compensatory decrease in synthesis of highly abundant liver enzymes. Next, in the following five sections, we show in steps that a decrease in liver B vitamin-related enzymes is a metabolic hallmark of SEKI and CRCA.

### Cofactors and enzymes exhibit a rough one-to-one stoichiometric ratio

As mentioned before, niacin and vitamin B6 decreased in the liver metabolome of SEKI and CRCA (Fig. 2b). Moreover, many hepatic NIA and B6-related enzymes were significantly downregulated in SEKI and CRCA (Fig. 4a). Since a previous study of *E. coli* data indicated a weak positive correlation between the abundance of metabolites and related proteins[40], we hypothesized a possible quantitative relation between these metabolites and enzymes. We then performed simple data retrieval and re-aggregation analysis to explore the relationship. For a metabolite from the mouse metabolome, we retrieved the names of all enzymes linked to that metabolite in the UniProt database, and then re-aggregated their reported amounts in mouse liver[41]. As shown in a previous study analyzing *E. coli* data, plotting the abundance of metabolites and related enzymes indicated a weak positive correlation between them (Fig. 4b)[40]. Another mouse liver quantitative proteomic dataset confirmed a similar quantitative relationship between metabolites and their corresponding enzymes (Fig. 4c)[42]. Furthermore, we found a steeper line between metabolites related to cofactors, including SAM and B vitamins, and cofactor-related enzymes (Fig. 4b, c).

To further investigate the relationship between cofactor-related metabolites, including SAM and B vitamins, and associated enzymes, we generated numerical biological objects (hereafter objects) by annotation-based data retrieval and re-aggregation. Specifically, we conducted a cofactor-related object-based analysis. First, we introduced a symbol for each cofactor object: SAM, SAM; B1, vitamin B1 (thiamine); B2, vitamin B2 (riboflavin); NIA, niacin including NAD; PA, pantothenic acid; B6, vitamin B6 (pyridoxine); Biotin, biotin; Folate, folate; B12, vitamin B12 (cobalamin) (Table S5). Next, we re-aggregated amounts of object-related metabolites and enzymes and then allocated the two values to each object. The cofactor-related objects showed a nearly linear correlation between metabolites and enzymes (Fig. 4d), indicating a strong quantitative relation between them in mouse liver.

A food composition database provides reliable concentrations of all B vitamins in foods. In addition, highly quantitative proteome datasets are available[42]. Plotting B-vitamin objects for three organisms from these public data revealed clear linear regression (Fig. 4d), confirming the linear relationship between B vitamins and associated enzymes across species. Each B-vitamin object had two attributes: metabolite and enzyme concentrations. The median molar ratio of B vitamin-related metabolites and enzymes fell between 0.1 and 10 in each species, and the average mean molar ratio from the four species converged close to one for each B vitamin except B12 (Fig. 4e). This result suggests a rough one-to-one stoichiometric ratio between cofactors and their related enzymes or a hidden symmetry between B vitamins and associated enzymes across four species (Fig. 4f).

How would such simple stoichiometry emerge between cofactor-related metabolites and their associated proteins? We speculate that this relationship may result from the fact that many enzyme

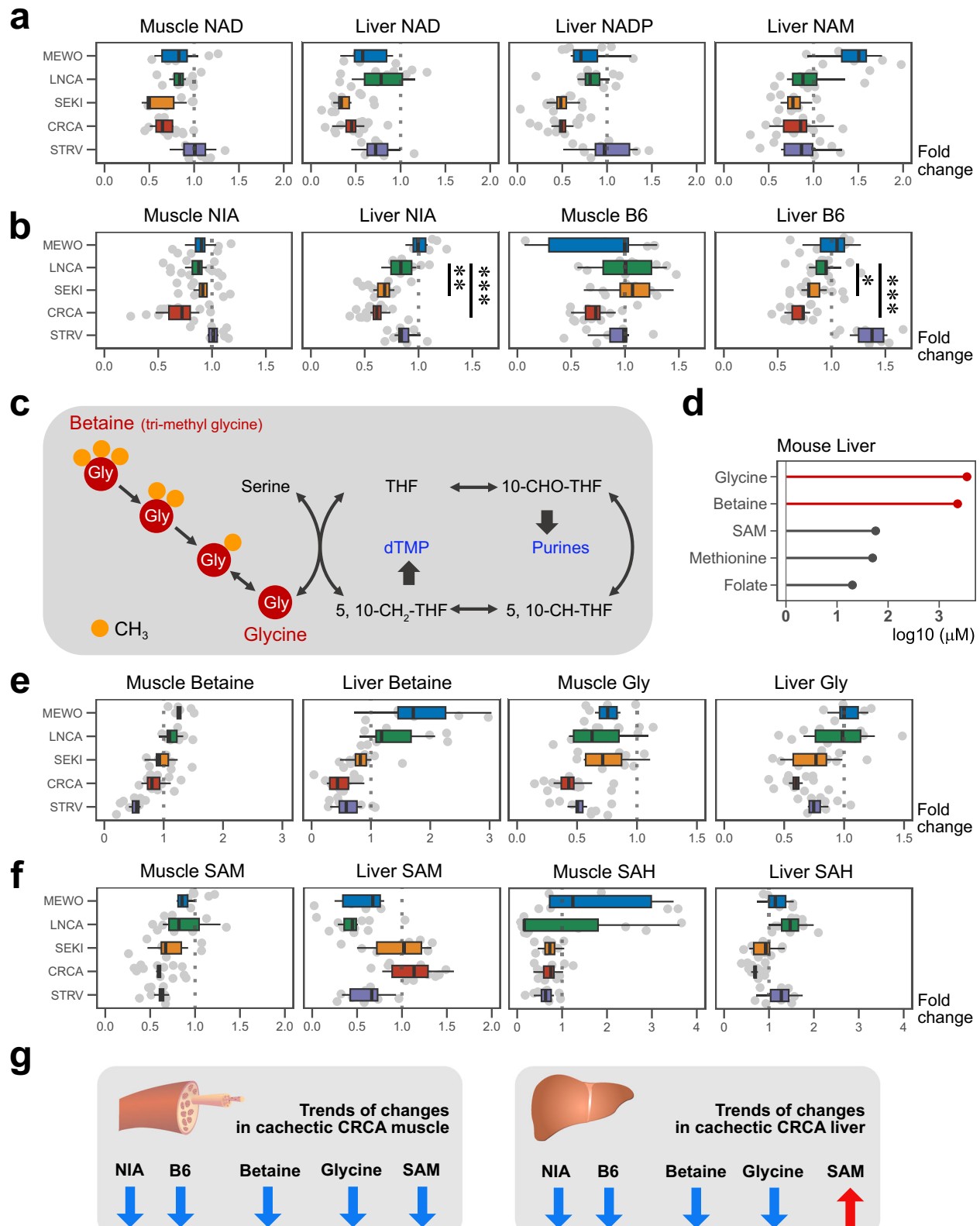

monomers have only one or two binding sites for metabolites. Rossmann fold structure, Rossmann fold-like structure, minimal Rossmann-like motif, and other related fold structures (hereafter collectively referred to as Rossmann-like domains; RLDs) are secondary protein structures commonly observed in various species from bacteria to humans[43,44]. The most well-characterized function of RLD is to bind nucleotide-based cofactors and metabolites such as NAD, FAD, and

SAM, and proteins with RLD reportedly covering ~38% of KEGG reference pathways[45]. For example, lactate dehydrogenase A (LDHA) and nicotinamide N-methyltransferase (NNMT) monomers have one RLD that captures one NAD molecule and one SAM molecule, respectively[43,46], and their RLDs comprise a large portion of these enzymes (Fig. 4g). RLDs of many other bacterial, yeast, and human enzymes also represent more than 50% of their total sequence lengths

**Fig. 2 | Highly abundant glycine and betaine are reduced, but low abundance SAM is maintained in SEKI and CRCA livers. a** Fold changes in NAD-related metabolites with respect to control mice. NAM, nicotinamide. **b** Fold changes of niacin (NIA) and vitamin B6 (B6). Dunnett's 2-sided test for LNCA (control group, $N = 11$ mice), SEKI ($N = 11$ mice), and CRCA ($N = 13$ mice) (Table S4), *$P < 0.05$, **$P < 0.01$, ***$P < 0.001$. Adjusted $P$-values (Dunnett's 2-sided test); Liver NIA, LNCA vs SEKI, $1.15 \times 10^{-3}$; Liver NIA, LNCA vs CRCA, $4.63 \times 10^{-6}$; Liver B6, LNCA vs SEKI, $1.98 \times 10^{-2}$; Liver B6, LNCA vs CRCA, $9.55 \times 10^{-8}$. **c** Schematic of the glycine-centered C1 pathway. THF tetrahydrofolate. **d** Mean concentrations of key C1 metabolites based on pooled C57BL/6N mouse liver data except folate[76]. **e** Fold changes in betaine and glycine with respect to control mice. **f** Fold changes in SAM and SAH. **g** Schematic summary of CRCA metabolome data. In the boxplots of Fig. 2, the middle line (median), the upper hinge (Q3, the third quantile), the lower hinge (Q1, the first quantile), the upper whisker ($1.5 \times (Q3 - Q1)$), and the lower whisker ($1.5 \times (Q3 - Q1)$). Muscle metabolome; MEWO, $N = 6$ mice; LNCA, $N = 11$ mice; SEKI, $N = 8$ mice; CRCA, $N = 13$ mice; STRV, $N = 8$ mice. Liver metabolome; MEWO, $N = 6$ mice; LNCA, $N = 11$ mice; SEKI, $N = 11$ mice; CRCA, $N = 13$ mice; STRV, $N = 8$ mice. Source data are provided as a Source Data file.

(the median total RLD sequence length; *E. coli*, 235 amino acids; *S. cerevisiae*, 264 amino acids; *H. sapiens*, 242 amino acids) (Fig. 4h)[45]. These observations suggest size requirements for RLDs to hold relatively large biomolecules, such as NAD (Molecular weight (MW), 664.433), FAD (MW, 789.5497) and SAM (MW, 398.4374), in the correct position to function as cofactors. Counting the number of RLDs per monomer revealed that ~90% of RLD enzymes from bacteria to humans carry only one or two RLDs (Fig. 4i). We speculate that numerical data structures found in this study, including symmetry and linearity, may reflect the constraint imposed by the number of metabolite binding sites per protein chain of the enzyme.

## A linear decrease of metabolites and enzymes related to NIA or B6 occurs in severe cachectic liver

Since it is plausible that linear systems exhibit a linear response to a disturbance[47], we examined whether the relationship between B vitamins and related enzymes exhibits linearity in livers of cancer-bearing mice (Fig. S6a). As expected, the fold changes in NIA-related enzymes decreased linearly with those of NIA-related metabolite levels (Fig. S6b). We observed a similar linear decrease in B6-related metabolites and enzymes. However, a clear linear response was not found between SAM-related metabolites and SAM-related enzymes that catalyze methylation, polyamine synthesis, and methionine processing. We then found that highly abundant NIA- and B6-related enzymes tend to act on metabolites present in high concentrations (Fig. S6c). This tendency was less clear for ATP- and SAM-related enzymes. This strong connectivity between highly expressed NIA- and B6-related enzymes and abundant metabolites may underlie the clear linear responses (Fig. S6b).

## Decreased liver B vitamin-related enzymes in SEKI and CRCA

To further examine the behavior of B vitamin-related enzymes in cancer cachexia, we performed a coarse-grained binary classification, a technique for analyzing proteome data[48,49]. First, we produced three mutually exclusive groups, a B vitamin-related enzyme group (B vitamin enzymes), a non-B vitamin enzyme group (other enzymes), and a non-enzyme protein group (non-enzyme proteins), according to UniProt annotations (Fig. S7a). According to public data, the median protein concentration in the B vitamin enzyme group was higher than in the other two groups (Fig. S7b). Next, we classified proteome data into two high and low groups according to the mean of protein concentrations, and also classified them into up and down groups based on their fold changes. B-vitamin enzymes were clustered more in the high and down category (greatly diminished expression) than other enzymes or non-enzymatic proteins in SEKI and CRCA (Fig. S7c). These results imply that hepatic B-vitamin enzymes are highly expressed in healthy conditions and exhibit an abundance-dependent decline as a group in cancer cachexia.

Then, we regarded slope as an indicator of abundance-dependent downregulation in the concentration/fold change plots (Fig. S7d). Notably, the B vitamin-related enzyme group showed a steeper downward slope than other enzymes or non-enzyme proteins (Fig. S7e). We also observed a correlation between SEKI and CRCA slopes (Fig. S7f), suggesting that SEKI and CRCA livers developed similar proteomic changes. Next, we obtained transcriptomic data from CRCA liver and plotted the fold changes of mRNAs against concentrations of their corresponding proteins (Fig. S7g). Again, the B vitamin-related enzyme group showed a steeper downward slope than other enzymes or non-enzyme proteins. In addition, the downward slopes of the transcriptome and proteome were correlated (Fig. S7h), suggesting that the abundance-dependent decrease in B vitamin-related enzymes in severe cachectic liver is regulated at the mRNA level.

To further explore the decline of B vitamin-related enzymes in cachectic liver, we focused on the four major B-vitamin enzymes, NIA, B6, PA, and B2 enzymes, which were well quantified in our proteome datasets, in addition to ATP enzymes as a reference. Enzymes related to NIA, B6, and PA, but not those related to B2 and ATP, showed an abundance-dependent declining trend in SEKI and CRCA livers (Fig. 5a). There seemed to be no substantial sex-related differences in collective behaviors of these enzymes in SEKI and CRCA livers (Fig. S8a, b). Transcriptomic data of CRCA livers also showed that the NIA-, B6-, and PA-related enzyme groups had steeper downward slopes than B2- or ATP-related enzymes, or all enzymes (Fig. 5b).

To examine shared features of NIA-, B6-, and PA-related enzymes, we performed binary classification of liver enzymes: enzymes in the highest quartile of concentration of liver proteins in public proteomic data were defined as high proteins and all others as non-high proteins. (Fig. 5c). The Liver B6-, NIA-, and PA-related enzyme groups contain more high enzymes, whereas peaks of the liver B2- and ATP-related enzyme groups were in the non-high zones. This result corresponds to the abundance-dependent decline of B6-, NIA- and PA-related enzymes, but not B2- and ATP-related enzymes in cancer cachexia (Fig. 5a). Furthermore, multiple regression analyses suggested that fold changes in liver enzymes are correlated with their relationships to NIA, B6, or PA cofactors, in SEKI and CRCA (Fig. 5d). These results suggest that types of cofactors may underlie proteomic changes in liver enzymes associated with cancer cachexia, and also imply that behaviors of liver enzymes influenced by their cofactors and abundance are involved in systemic metabolic changes in severe cancer cachexia (Fig. 5e).

## Severe cachectic liver shows a decrease in Gly-related enzymes, but not in SAM-related enzymes

To investigate whether and how these abundance-dependent proteomic changes in SEKI and CRCA livers influence C1 metabolism, we split C1-related enzymes into two sub-groups based on their dependence on B vitamins (Fig. 6a). Levels of B vitamin-dependent C1 enzymes generally declined, but those of B vitamin-independent C1 enzymes tended to increase in SEKI and CRCA livers. Upregulated C1 enzymes in the severe cachectic livers included fewer B vitamin-dependent and Gly-related enzymes that catalyze Gly as a substrate and more SAM-related enzymes (Fig. 6b).

SEKI and CRCA livers showed concentration-weighted downregulation of Gly-related enzymes, but not SAM-related enzymes (Fig. 6c). Transcriptomic data supported this proteomic finding (Fig. 6d). According to public proteome data, most Gly-related enzymes are present at higher concentrations, whereas many SAM-related enzymes occur at lower concentrations in liver (Fig. 6e). Levels

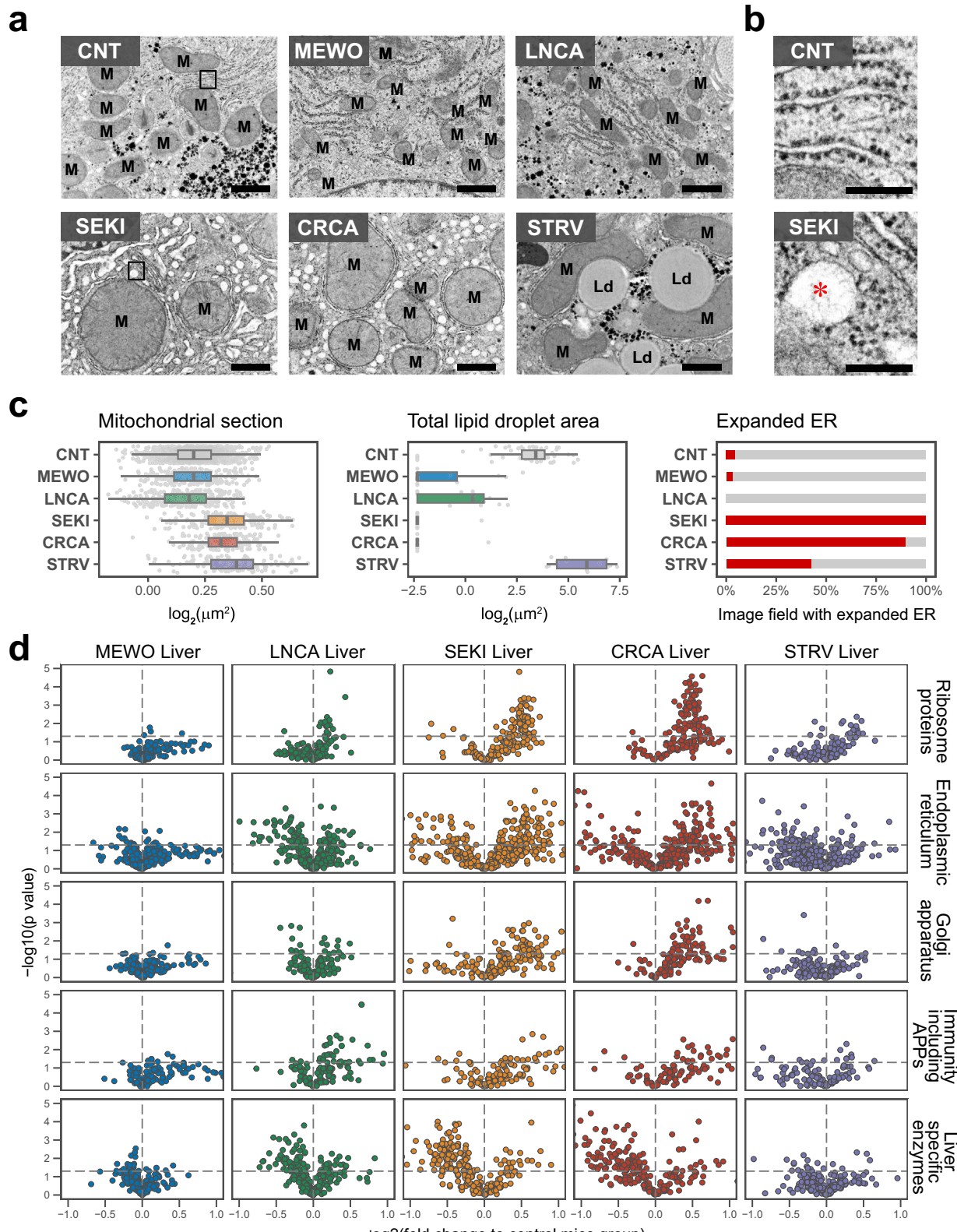

of hepatic Gly-related enzymes were positively correlated with those of NIA- or B6-related metabolites and Gly (Fig. 6f). Collectively, these lines of circumstantial evidence led us to propose the following hypothesis: levels of B vitamin-dependent Gly-related enzymes are reduced in severe cachectic livers in parallel with the decrease in Gly, but B vitamin-independent SAM-related enzymes and SAM are maintained (Fig. 6g).

## Systemic decline of protein lysine malonylation in SEKI and CRCA

Pantothenic acid (PA), vitamin B5, is a major component of coenzyme A (CoA), an essential intermediate metabolite used in intracellular exchange of acyl groups[50]. According to the UniProt database, PA (CoA)-related enzymes comprise almost half of acyl-related enzymes (Fig. S9a). Reactive acyl-CoA molecules, including acetyl-CoA,

**Fig. 3 | SEKI and CRCA livers show structural and proteomic changes, accompanied by overproduction of acute-phase proteins. a** Representative transmission electron microscopic (EM) images of hepatocytes. CNT, control nude mouse; M, mitochondria; LD, lipid droplet. Bars, 1 μm. CNT, $N = 2$ mice; MEWO, $N = 1$ mouse; LNCA, $N = 1$ mouse; SEKI, $N = 1$ mouse; CRCA, $N = 1$ mouse; STRV, $N = 1$ mouse. **b** Enlarged views of CNT and SEKI showing rough endoplasmic reticula (ER). Red asterisk, expanded ER lumen. Bars, 250 nm. **c** Quantified morphological changes. Mitochondrial section, mitochondrial cross section area. Total lipid droplet area, total lipid droplet area of one imaging area (area size, 581.92 μm²) Expanded ER, percentage of image fields with expanded ER. CNT, EM images from 5 control mice. CNT, $N = 5$ mice; MEWO, $N = 1$ mouse; LNCA, $N = 1$ mouse; SEKI, $N = 1$ mouse; CRCA, $N = 1$ mouse; STRV, $N = 1$ mouse. Boxplot style, the middle line (median), the upper hinge (Q3, the third quantile), the lower hinge (Q1, the first quantile), the upper whisker ($1.5 \times (Q3 - Q1)$), the lower whisker ($1.5 \times (Q3 - Q1)$). Details of electron micrograph quantification are described in the Methods section (Statistics and Reproducibility). **d** Keyword-based volcano plots of liver proteomic data. MEWO, $N = 3$ mice; LNCA, $N = 3$ mice; SEKI, $N = 3$ mice; CRCA, $N = 3$ mice; STRV, $N = 2$ mice. Horizontal dashed lines show where $P = 0.05$ (Welch's 2-sided $t$-test). APPs, acute-phase proteins. Plots are limited to a range of −1 to 1 on the X-axis and 0 to 5 on the Y-axis. Source data are provided as a Source Data file.

succinyl-CoA, and malonyl-CoA, often modify highly nucleophilic lysine (K) residues of proteins (Fig. S9b)[51]. The massive decline of PA-related enzymes in cachectic liver (Fig. 5a) led us to analyze the acetylation, succinylation, and malonylation status of liver proteins by LC-MS/MS-based post-translational modification (PTM) profiling. Volcano plots demonstrated alterations of protein acylation in muscle and liver of SEKI and CRCA mice (Fig. S9c).

Lysine acylation by reactive acyl-CoA often proceeds non-enzymatically; therefore, various acyl-CoA molecules can compete for a single lysine site (Fig. 7a)[51]. To explore possible involvement of such interference, we determined the lysine sites that overlapped among acetylation, malonylation, and succinylation (Fig. 7b). We then recognized a V-shaped pattern on the graph, indicating that lysine malonylation for shared sites is reduced in cachectic muscle and liver, compared with acetylation or succinylation (Fig. 7c), possibly due to competition for shared lysine sites. Notably, western blot analysis confirmed a significant systemic decline of protein malonylation in SEKI and CRCA mice, but not in STRV mice (Fig. 7d).

We observed an abundance-dependent decline of acyl-related, acetylation-related, and succinylation-related enzymes in SEKI and CRCA livers (Fig. S9d). Although the downward trend of malonylation-related enzymes was not evident because of the small number of malonylation-related proteins identified in this study, transcriptomic data suggested an abundance-dependent decline of malonylation-related enzymes (Fig. 7e). Furthermore, public proteomic data show that most malonylation-related enzymes are highly abundant in the liver and are B vitamin-related enzymes (Fig. 7f, g). Taken together, our analysis of mouse models suggests that metabolomic and proteomic changes characteristic of SEKI and CRCA, including those in C1 metabolism and protein acylation, are all related to the decrease in B vitamin-related enzymes (Fig. 7h). We propose this feature, centered on B vitamin-related enzymes, as a metabolic hallmark of severe cancer cachexia.

### Blood samples from severe cachexia models and advanced gastric cancer cachexia patients show decreases in C1-related metabolites and malonylation

To address whether the above-mentioned candidate hallmark of severe cancer cachexia is reflected in the blood and to assess its clinical relevance, we performed metabolomic analyses using blood samples from CRCA mice and from gastric cancer patients, who frequently develop severe cachexia (Table S6)[52]. We listed blood metabolites that were significantly altered in CRCA mice ($N = 13$) compared with control C57BL/6N mice ($N = 21$), and in advanced gastric cancer patients diagnosed with cachexia ($N = 25$) compared with early gastric cancer patients without cachexia ($N = 14$; Table S6). We then compared the mouse and human short lists (Fig. 8a). As a result, we identified 15 metabolites that exhibited similar changes both in CRCA mice ($N = 13$) and advanced gastric cancer patients diagnosed with cachexia, according to criteria of Fearon et al. ($N = 25$) (Fig. 8a)[1]. The 15 metabolites included five C1-related metabolites, betaine, serine (Ser), threonine (Thr), choline, and guanidinoacetate (GAA) and an NAD-related amino acid, tryptophan (Trp) (Fig. 8a)[34,53]. ROC curve analysis

suggested that a combination of choline and Trp predicts cachexia more successfully than CRP and albumin (Fig. S10).

To further evaluate the clinical relevance of our metabolomic data, we examined the relationship between the six metabolites and the Glasgow prognostic score (GPS)[54]. GPS is a clinical scoring system for cancer patients, defined by two liver-derived acute-phase proteins (APPs), C-reactive protein (CRP; positive APP) and albumin (negative APP). Significantly, GPS correlates positively with the severity of cachexia and negatively with the prognosis of cancer patients[54]. Notably, we observed significant negative relations between GPS and the five C1-related metabolites and the NAD-related metabolite (Fig. 8b), supporting an association between gastric cancer cachexia and these six metabolites.

SEKI and CRCA livers exhibited significant changes in protein acylation (Fig. S9c). The liver secretes many blood proteins (Fig. 8c), including albumin, transferrin, APPs, and apolipoproteins. We thus examined malonylation and acetylation of blood proteins from SEKI and CRCA mice (Fig. 8d). We observed that liver-derived plasma proteins exhibited decreasing trends for malonylation and acetylation in both SEKI and CRCA (Fig. 8e). Furthermore, analysis of blood samples from advanced gastric cancer cachexia patients ($N = 4$) and early gastric cancer patients without cachexia ($N = 6$) revealed that most malonylated and acetylated proteins commonly detected in mouse and human plasma are primarily liver-derived (Fig. S11a, b) and that some cases show a tendency toward decreased malonylation and acetylation, similar to SEKI and CRCA (Fig. 8e). Liver-derived apolipoproteins tended to lose malonylation in the blood of three of the four patients with advanced gastric cancer tested, and decreased apolipoprotein acetylation was observed in all four patients, highlighting a link between gastric cancer cachexia and serum protein acylation.

## Discussion

This study revealed the landscape of systemic metabolic changes in cancer cachexia and identified a decrease in B vitamin-related liver enzymes as its hallmark, providing a metabolic framework to understand its pathophysiology (Fig. 8f).

Skeletal muscle wasting in cancer cachexia is a critical medical problem. The link between muscle wasting and liver abnormalities has already been suggested by studies of non-alcoholic fatty liver disease (NAFLD) and cirrhosis[55–57]. A recent study suggests that skeletal muscle may not be a storage tissue for amino acids, but rather a consumer of amino acids such as serine and glycine supplied by the liver and kidney[58]. Furthermore, the liver synthesizes nicotinamide (NAD precursor) and supplies it to peripheral tissues such as skeletal muscle[25]. Notably, cachectic mice had smaller liver nicotinamide and NAD pools (Figs. 2a, 8f). These findings lead us to speculate that cachectic liver may not be able to secrete a sufficient amount of metabolites that are essential to maintain skeletal muscle. A complete analysis of the quantitative relationship between skeletal muscle mass and the hepatic capacity to secrete important metabolites will be crucial for better understanding the mechanism of muscle loss associated with cancer cachexia.

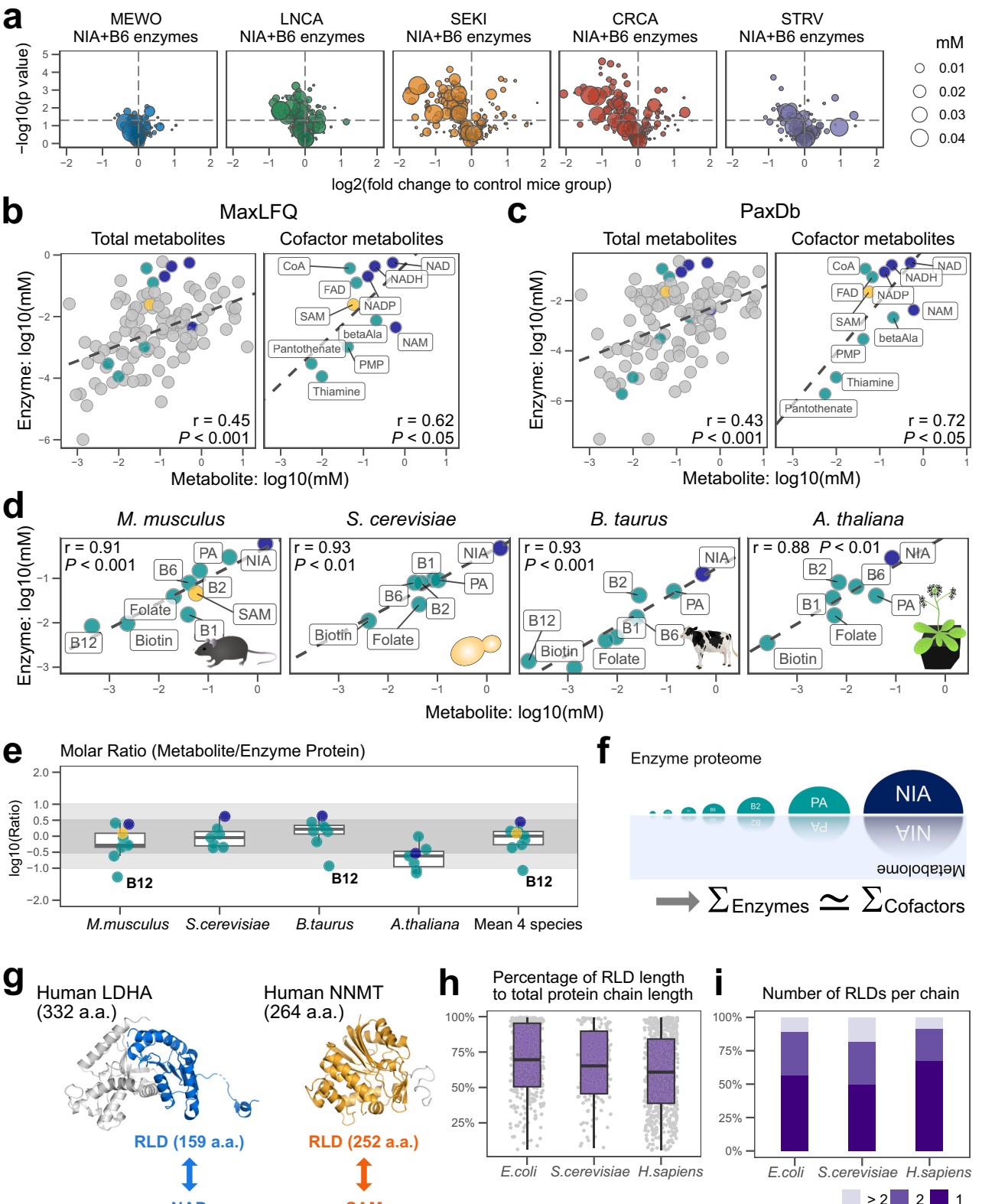

Our proteome data indicate enhanced ribosome biogenesis in SEKI and CRCA livers, probably associated with acute-phase proteins (Figs. S1e, 3a, d). Ribosome biogenesis imposes high metabolic costs and requires large amounts of nucleic acids synthesized with intermediate metabolites provided by the C1 pathway (Fig. 2c)[59]. The average hepatocyte has 6 million ribosomes[60], and diurnal oscillations cause fluctuations of hepatic RNA and protein masses and even liver weight[61]. Thus, enhanced ribosome biogenesis in cachectic liver may be one of the driving forces that transform the landscape of liver metabolism (Fig. 8f). Several clinical studies have reported that the concentration of C-reactive protein (CRP), a liver-derived acute-phase protein, correlates with the severity of cachexia and prognosis of cancer patients[16,17,54]. The adverse influence of hyperactivated ribosome biogenesis on liver metabolism will be the subject of a future

**Fig. 4 | Cofactors and enzymes exhibit a rough one-to-one stoichiometric ratio.**
**a** Keyword-based volcano plots of liver proteomic data (Table S5). NIA + B6, NIA- and B6-related enzymes. Bubble size, estimated enzyme concentrations from published proteomic data[74]. Calculation of the protein concentration estimates is described in the Methods section (Methods, Estimation of protein concentration). Horizontal dashed lines, $P = 0.05$ (Welch's 2-sided $t$-test). Plots are limited to a range of −2 to 2 on the X-axis and 0 to 5 on the Y-axis. MEWO, $N = 3$ mice; LNCA, $N = 3$ mice; SEKI, $N = 3$ mice; CRCA, $N = 3$ mice; STRV, $N = 2$ mice. **b** Scatterplots of metabolites and related enzyme proteins in mouse liver using mouse liver MaxLFQ data. Mouse liver metabolome data; C57BL/6N, $N = 59$ mice. Dark blue circles, NAD-related metabolites; cyan circles, other B vitamin-related metabolites; orange circles, SAM-related metabolites. Dashed lines represent linear regression. r, Pearson's correlation coefficient of log transformed data. Non-adjusted $P$-value (Pearson's correlation); total metabolites, $2.24 \times 10^{-7}$; cofactor metabolites, 0.0425. **c** Scatterplots of metabolites and related enzyme proteins in mouse liver using mouse liver PaxDb data[75]. Mouse liver metabolome data; C57BL/6N, $N = 59$ mice. r, Pearson's correlation coefficient of log transformed data. Non-adjusted $P$-value (Pearson's correlation); total metabolites, $1.54 \times 10^{-6}$; cofactor metabolites, 0.0125. **d** Linearity among B-vitamin-objects on log-log plots. Dashed lines represent linear regression. Protein data were integrated from PaxDb datasets. Mouse B vitamins and SAM, mouse liver metabolome data generated in this study (Mouse metabolome data; C57BL/6N, $N = 59$ mice). Mouse folate, biotin, and B12 values were collected from previous reports[79,80,85]. B vitamin data of *S. cerevisiae*, *B. taurus*, and *A. thaliana*, Standard Tables of Food Composition in Japan, 2015. Details of estimation of B vitamin concentrations from CE-MS data or food datasets are described in the Methods section. **r** Pearson's correlation coefficient of log transformed data. Non-adjusted $P$-values (Pearson's correlation); *M. musculus*, $7.56 \times 10^{-4}$; *S. cerevisiae*, 0.00164; *B. taurus*, $7.23 \times 10^{-4}$; *A. thaliana*, 0.00834. **e** The molar ratio of total metabolite abundance to total enzyme protein abundance. The gray zone is within ±1 in the base-10 representation of the ratio. Orange circle, SAM; Dark blue circles, NIA; cyan circles, other members of B-vitamin-objects. B12 metabolites are not reported in the yeast and plant data, and their B12 values are missing. Boxplot style, the middle line (median), the upper hinge (Q3, the third quantile), the lower hinge (Q1, the first quantile), the upper whisker $(1.5 \times (Q3 - Q1))$, the lower whisker $(1.5 \times (Q3 - Q1))$. Note that **e** is based on the same data used in **d**. **f** Conceptual diagram of symmetry of B vitamin-related objects across the metabolome and enzyme proteome. Σ data aggregation, ≃ approximately equal. **g** Structures of human LDHA and NNMT, based on publicly accessible data[37,45]. RLD (Rossmann fold domain) colored in blue or orange. a.a. amino acids. **h** Box plots of RLD length as a percentage of total protein chain length of enzyme, based on publicly accessible data[37,45]. *E. coli*, 259 enzymes; *S. cerevisiae*, 108 enzymes; *H. sapiens*, 465 enzymes. Boxplot style, the middle line (median), the upper hinge (Q3, the third quantile), the lower hinge (Q1, the first quantile), the upper whisker $(1.5 \times (Q3 - Q1))$, the lower whisker $(1.5 \times (Q3 - Q1))$. **i** Bar graph of the number of RLD per RLD-containing enzyme, based on publicly accessible data[37,45]. *E. coli*, 259 enzymes; *S. cerevisiae*, 108 enzymes; *H. sapiens*, 465 enzymes. > 2, more than two RLDs; 2, two RLD; 1, one RLD per single enzyme protein chain. Source data are provided as a Source Data file.

study, but taming an abnormal ribosomal system could be a novel therapeutic approach for cancer cachexia.

This study suggests candidate biomarkers for cancer cachexia: blood choline, tryptophan, and acylation profiles of blood proteins (Figs. 8, S10). Although the small number of clinical samples and analysis of a single cancer type are limitations of this study, our findings on choline and tryptophan agree with previous reports[62–64]. On the other hand, few reports have described the acylation status of blood proteins in cancer cachexia or other diseases. Although extensive validation is needed to establish clinically relevant biomarkers for cachexia, mapping and deciphering acylation patterns of blood proteins may identify potential candidate biomarkers.

In conclusion, our multiple 'omics' analysis of multiple mouse models reveals how cancer cachexia transforms the landscape of liver metabolism (Fig. 8f). Future mechanistic research on the individual metabolites, proteins, or PTMs addressed in this study may lead to novel prophylactic, therapeutic, or mitigative strategies for improving the quality of life of patients suffering from this illness.

## Methods

### Ethics statement

This research complies with all relevant ethical regulations in Japan. All animal experiments were performed according to protocols approved by the Animal Care and Use Committee of Aichi Cancer Center Research Institute (#27-9, #28-15, #30-3, #30-9, #31-14, and #R2-7). This study was also approved by the ethics committee of Aichi Cancer Center Hospital (No. 2015-2-13; No. 2019-1-540) and conducted following the Helsinki Declaration and Japanese Ethical Guidelines for Clinical Studies Involving Human Subjects.

### Animal studies

BALB/c nu/nu (BALB/cAJcl nu/nu) and C57BL/6N (C57BL/6NJcl) mice were purchased from CLEA Japan (Tokyo, Japan) and were acclimated for at least 1 week before any experimental procedures. *K-ras*[LSL-G12D/+] (Stock No: 008179; RRID: IMSR_JAX:008179) and *p53*[lox/lox] (Stock No: 008462; RRID: IMSR_JAX:008462) mice were obtained from the Jackson Laboratory. After crossing with C57BL/6N mice for several generations, *K-ras*[LSL-G12D/+]; *p53*[lox/lox] (KP) mice (referred to as LNCA in this study) were produced by crossing mice with *K-ras*[LSL-G12D/+] and *p53*[lox/lox] mice[12]. Details of generation of *cis-Apc*[+/Δ716]/*Smad4*[+/-] mice (referred to as CRCA in this study) have been reported previously[14]. *cis-Apc*[+/Δ716]/

*Smad4*[+/-] male mice were crossed with C57BL/6N female mice for more than twenty generations. Due to loss of heterozygosity, *cis-Apc*[+/Δ716]/ *Smad4*[+/-] mice spontaneously develop multiple invasive intestinal carcinomas and generally become moribund within 4 months of age. For metabolomic and proteomic analysis, we mainly used female *cis-Apc*[+/ Δ716]/*Smad4*[+/-] mice and their female littermates.

Mice were housed in a specific-pathogen-free facility, kept at room temperature with standard day-night cycles, and provided with commercial radiation-sterilized laboratory chow (CLEA Rodent Diet CE-2; CE-2, 30 KGY, 10 kg; CLEA Japan, Tokyo, Japan) and autoclaved tap water *ad libitum*. The ingredient and compositional data of non-radiation sterilized CE-2 chow are available at the following URL: https://www.clea-japan.com/en/products/general_diet/ item_d0030. Body and food pellet weights were measured with an electronic balance with a bucket for small animals (UX2200H, Shimadzu). We calculated daily food intake from the simple daily change in the weight of pellets in the wire-bar hopper on the cage lid. Adult 8-week-old female BALB/c nu/nu mice were subcutaneously injected with $1 \times 10^6$ MEWO or SEKI melanoma cells under anesthesia (medetomidine hydrochloride (0.75 mg/kg), midazolam (4 mg/kg), butorphanol tartrate (5 mg/kg)). Adenovirus-Cre (Ad5CMVCre, VVC-U of Iowa-5, University of Iowa) was delivered to 6–12-week-old KP mice using an intratracheal infection method under anesthesia as described above[12]. After injection of melanoma cells or inhalation of recombinant adenovirus, atipamezole hydrochloride (0.75 mg/kg) was immediately given as an anesthetic antagonist. In starvation experiments, adult 12-week-old C57BL/6N mice were deprived of laboratory chow for up to 48 h, but were allowed continuous access to water[57]. When establishing a starvation protocol to help characterize cancer cachexia, we compared 24-h and 48-h starvation as a pilot experiment. Some 24-h fasted mice maintained adequate muscle and white adipose tissue weights. Therefore, we selected the 48-h starvation model. In 5-Fluorouracil (5FU) studies, we treated 12-week-old female C57BL/6N mice with 5FU (100 mg/kg body weight, Cat#068-01401, FUJIFILM Wako, Osaka, Japan) once a day for 5 days. 5FU was dissolved in saline administrated by intraperitoneal injection. In FK866 studies, we treated young 6-week-old female C57BL/6N mice with FK866 (40 mg/kg body weight, S2799, Selleck) twice a day for 2 days. FK866 was suspended in 30% propylene glycol (Cat#164-04996, FUJIFILM Wako), 5% polyoxyethylene (20) sorbitan monooleate

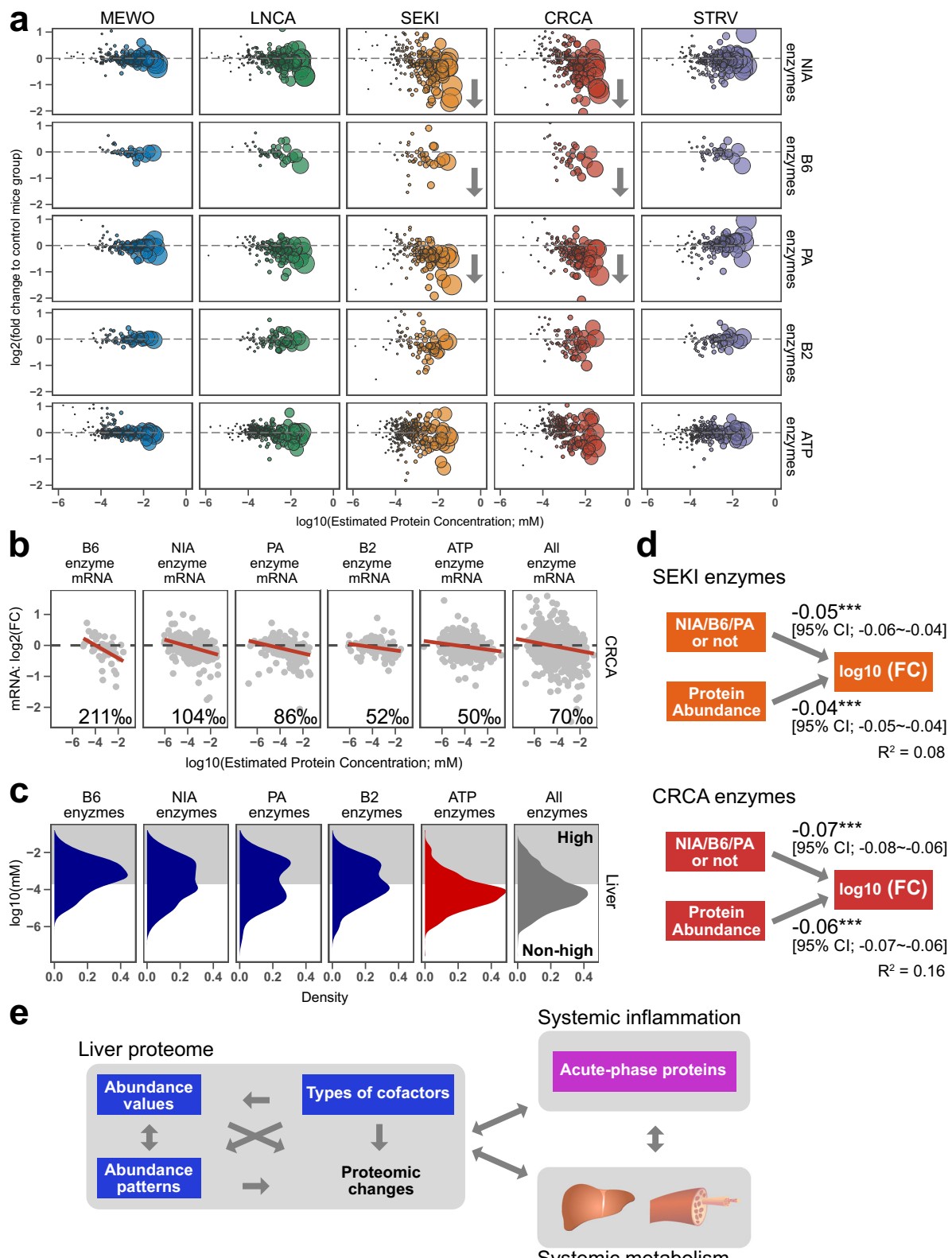

(Cat#169-04996, FUJIFILM Wako), 65% intravenous sugar solution and administered by intraperitoneal injection.

We conducted all the animal experimental activities, including monitoring the health and tumor burden of all the mice analyzed in this study, under protocols approved by the Animal Care and Use Committee of Aichi Cancer Center Research Institute (#27-9, #28-15, #30-3, #30-9, #31-14, and #R2-7). We monitored the size of SEKI and

MEWO xenograft using a caliper. To estimate the xenograft size, we used the formula: Tumor volume = ½ (length × width2). The xenografts in this study did not exceed the maximal tumor size (2 cm$^3$) allowed by the Animal Care and Use Committee of Aichi Cancer Center Research Institute. In this study, we used the following experimental model-based endpoints for valid scientific outcomes. SEKI xenograft mice and *cis-Apc$^{+/\Delta716}$/Smad4$^{+/-}$* mice reproducibly exhibited hunching

**Fig. 5 | Decreased liver B vitamin-related enzymes in SEKI and CRCA. a** Declines of vitamin B-related enzymes in SEKI and CRCA liver proteomes. Bubble size represents protein concentration. MEWO, $N = 3$ mice; LNCA, $N = 3$ mice; SEKI, $N = 3$ mice; CRCA, $N = 3$ mice; STRV, $N = 2$ mice. MaxLFQ values were used to estimate protein concentrations[41]. The calculation of the protein concentration estimates is described in the Methods section (Methods, Estimation of protein concentration). Fold changes are from TMT quantitative proteomic analysis. NIA, niacin related enzymes; B6, vitamin B6 related enzymes; PA, pantothenic acid related enzymes; B2, vitamin B2 related enzymes; ATP, ATP related enzymes. The plots are limited to a range of −6 to 0 on the X-axis and −2 to 1 on the Y-axis. **b** Scatterplots of changes (log2) in the CRCA liver transcriptome with published mouse protein abundance data on the X-axis. FC, fold change. Red solid lines represent linear regression lines. Transcriptome data; CRCA, $N = 4$ mice; C57BL/6N, $N = 4$ mice. Protein concentrations were estimated from MaxLFQ values of mouse liver by LC-MS. Facet number, depression slope (permille). **c** Distribution of B6-, NIA-, PA-, B2-, ATP-related and all enzymes in mouse hepatic proteome data. High concentrations (gray zone), protein concentrations above the highest quartile of the published hepatic proteome.

**d** Path diagrams of multiple regression analysis for SEKI and CRCA hepatic proteome data (SEKI, $N = 3$ mice; CRCA, $N = 3$ mice). NIA/B6/PA or not, a dummy variable, specifically, NIA-, B6-, PA-related enzymes (= 1) or other enzymes (= 0) (SEKI, 1259 enzymes; CRCA, 1008 enzymes); Protein abundance, protein concentrations estimated from MaxLFQ values of mouse liver by LC-MS; FC, fold change values from proteome data; unidirectional arrows, cause-and-effect relationships; numbers next to arrows, partial regression coefficients; asterisks next to partial regression coefficients, non-adjusted P-values (multiple regression analysis, ***$P < 0.001$); $R^2$, non-adjusted multiple R-squared; 95% CI, 95% confidential interval. Non-adjusted P-values (multiple regression analysis); SEKI, NIA/B6/PA or not, $8.68 \times 10^{-7}$; SEKI, protein abundance, $7.29 \times 10^{-16}$; CRCA, NIA/B6/PA or not, $5.46 \times 10^{-11}$; CRCA protein abundance, $2 \times 10^{-16}$. Generalized variance-inflation factor (GVIF); SEKI, NIA/B6/PA or not, 1.056; SEKI, protein abundance, 1.060; CRCA, NIA/B6/PA or not, 1.052; CRCA, protein abundance, 1.052. **e** Schematic of relationships between key factors, proposed mainly based on proteomic findings. Source data are provided as a Source Data file.

behavior[65]. We set the Sevcik hunching score of 4 as the humane endpoint for SEKI and $cis$-$Apc^{+/\Delta716}/Smad4^{+/-}$ mice. Specifically, mice scored as 4 had a severe rounded-back posture and whole-body piloerection, and little exploratory behavior[65]. As far as we observed, the most prominent physical finding of KP mice was labored breathing. KP mice were thus carefully monitored for respiration rate and euthanized if severe, persistent hyperventilation occurred[66]. MEWO xenograft mice did not manifest critically ill health status, even just before reaching the acceptable limit of tumor burden. Based on the assessment of superficial tumor size, MEWO mice were euthanized 3–10 weeks after injection.

We collected all mouse samples from ad-lib fed conditions, except the starvation group. To obtain mouse tissue samples for further analysis, both experimental and control mice were deeply anesthetized and euthanized between 2–4 pm on the same day, to the extent possible. After euthanasia, tissue samples were immediately collected, rinsed with 5% w/v mannitol solution (Cat#133-00845, FUJIFILM Wako) at least twice, snap-frozen in liquid nitrogen, and stored at −80 °C for further analysis. Mouse blood samples were collected by cardiac puncture under deep anesthesia, immediately followed by a secondary method of euthanasia. Mouse plasma was prepared by centrifugation from collected blood with a final 0.13 w/w% concentration of EDTA-2K.

In the B vitamin treatment experiment, 8-week-old female BALB/c nu/nu mice were randomly assigned to two groups, 6 days after inoculation with SEKI cells ($1.0 \times 10^6$ cells) and intraperitoneally injected with a clinical-grade B vitamin cocktail (Otsuka MV injection No.1 vial consisting solely of water-soluble vitamins, 3179513A1026, Otsuka Pharmaceutical Co., Ltd., Tokyo, Japan) or saline. A quarter-vial dose was administered daily to each mouse (thiamine chloride hydrochloride, 49 mg/kg; riboflavin sodium phosphate, 58 mg/kg; pyridoxine hydrochloride, 61 mg/kg; cyanocobalamin, 0.06 mg/kg; nicotinamide, 500 mg/kg; folic acid, 5 mg/kg; ascorbic acid, 1250 mg/kg; panthenol, 175 mg/kg; biotin, 0.75 mg/kg). Sample sizes were determined empirically (B vitamin treatment, $N = 12$ female mice; normal saline treatment, $N = 12$ female mice). Primary experimental endpoints and statistical methods were the same as in the glycine treatment experiment mentioned above.

In the glycine treatment experiment, 8-week-old female BALB/c nu/nu mice were randomly divided into two groups 5 days after inoculation of SEKI cells ($1.0 \times 10^6$ cells) and followed by glycine or saline treatment. 100 µL of glycine saline (glycine, 1000 mg/kg) or saline were administered intraperitoneally using a 29-gauge needle daily. Experiments were performed in duplicate, and data were pooled (glycine treatment, $N = 23$ female mice; normal saline treatment, $N = 24$ female mice). Sample sizes were determined empirically. The primary experimental outcome was survival time after inoculation. Kaplan-Meier survival analysis and log-rank tests were performed. The date of

euthanasia relative to inoculation date was used for Kaplan-Meier analysis. A value of $P < 0.05$ was considered significant. To examine the effect of glycine administration on SEKI xenografts, we also conducted an experiment in which all mice were euthanized and examined 22 days after inoculation with SEKI cells and 17 days after the start of high-dose glycine administration (glycine, 1000 mg/kg; glycine treatment, $N = 8$ female mice; normal saline treatment, $N = 8$ female mice).

### Cell Lines and cell culture conditions
Human melanoma cell lines, mycoplasma-free MEWO (JCRB0066 [https://cellbank.nibiohn.go.jp/~cellbank/en/search_res_det.cgi?ID = 241]; RRID: CVCL_0445) and mycoplasma-free SEKI (JCRB1041 [https://cellbank.nibiohn.go.jp/~cellbank/en/search_res_det.cgi?ID = 2416]; RRID: CVCL_3162), were obtained from JCRB Cell Bank (Osaka, Japan) and cultured in RPMI-1640 medium (30263-95; Nacalai Tesque, Kyoto, Japan) with 10% fetal bovine serum (Sigma). We maintained both cell lines in standard tissue culture conditions of 37 °C and 5% $CO_2$. MEWO and SEKI are not listed on the Register of Misidentified Cell Lines (v12, 2023) provided by International Cell Line Authentication Committee (ICLAC [https://iclac.org/databases/cross-contaminations/]). The authenticities and mycoplasma contamination of these cells were not tested in our laboratory.

### Human studies
We retrospectively analyzed frozen stocked human residual blood samples from Japanese patients with gastric cancer at Aichi Cancer Center Hospital, obtained written informed consent, or granted the opt-out of informed consent. The opt-out approach received approval from the institutional review board of Aichi Cancer Center Hospital, taking into account the retrospective observational nature of the study and the absence of any participant risk (No. 2019-1-540). Diagnosis and clinical staging of gastric cancer was based on Japanese gastric cancer treatment guidelines 2014. We classified early gastric cancer and advanced cancer according to the clinical stage at the time of blood sampling (total $n = 57$; early gastric cancer, stage I and II, $n = 16$; advanced gastric cancer, stage III and IV, $n = 41$) and stratified patients into three groups using the classical Glasgow Prognostic Score (GPS = 0, $n = 20$; GPS = 1, $n = 10$; GPS = 2, $n = 27$)[54]. According to diagnostic criteria for cancer cachexia by Fearon et al.[1], we also classified 30 advanced gastric cancer patients whose body weight could be tracked in their medical records for ~6 months from the time of blood sampling into the cachexia ($N = 25$) and non-cachexia groups ($N = 5$). Patients with early gastric cancer were diagnosed with cancer cachexia if they self-reported body weight that matched Fearon et al.'s diagnostic criteria (cachexia, $N = 2$; non-cachexia, $N = 14$). Presence or absence of gastrointestinal obstruction was determined by reviewing medical records.

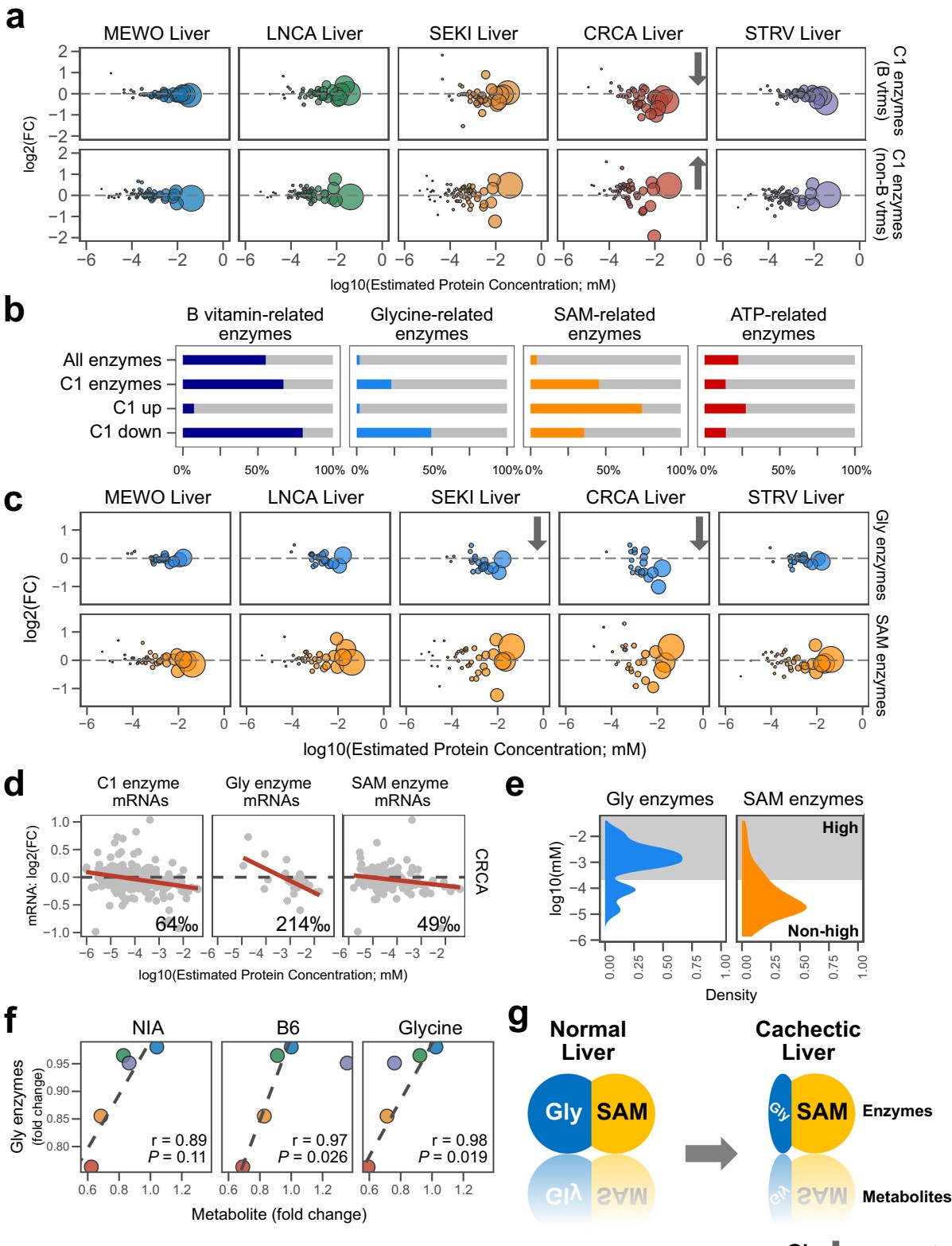

## Real-time RT-PCR analysis

According to the manufacturer's instructions, total RNA was isolated from muscle samples using an RNeasy Fibrous Tissue Mini Kit (QIAGEN) with on-column RNase-free DNase I (QIAGEN) treatment. A High-Capacity cDNA Reverse Transcription Kit (Life Technologies) was used for synthesis of complementary DNA from total RNA.

mRNA levels of *Trim63*, *Fbxo32*, *Foxo1*, and *Gapdh* were quantified with TaqMan Gene Expression assays (Mm01185221_m1, Mm00499523_m1, Mm00490671_m1, and Mm99999915_g1, respectively) from Thermo Fisher Scientific. *Gapdh* was used as an internal control to normalize expression levels of *Trim63*, *Fbxo32*, and *Foxo1* by the comparative $2^{-\Delta\Delta CT}$ method.

**Fig. 6 | SEKI and CRCA livers show a decrease in Gly-related enzymes, but not in SAM-related enzymes. a** Declines of C1 enzymes related to B vitamins in SEKI and CRCA liver proteomes. Bubble size represents protein concentration estimated from published proteome data. The calculation of the protein concentration estimates is described in the Methods section (Methods, Estimation of protein concentration). Changes (FC) are from TMT quantitative proteomic analysis. B vtms, C1 enzymes related to B vitamins; non-B vtms, C1 enzymes unrelated to B vitamins. Plots are limited to a range of −6 to 0 on the X-axis and −2 to 2 on the Y-axis. MEWO, $N = 3$ mice; LNCA, $N = 3$ mice; SEKI, $N = 3$ mice; CRCA, $N = 3$ mice; STRV, $N = 2$ mice. **b** Percentage of amounts of B vitamin-related enzymes (dark bule), Gly-related enzymes (sky blue), SAM-related enzymes (orange), and ATP-related enzymes (red) to total amounts of specified enzyme groups in liver. C1 UP, upregulated C1 enzymes in livers of SEKI and CRCA; C1 DOWN, downregulated C1 enzymes. **c** Alterations of Gly and SAM enzymes in livers of five experimental models. Plots

are limited to a range of −6 to 0 on the X-axis and −1.5 to 1.5 on the Y-axis. MEWO, $N = 3$ mice; LNCA, $N = 3$ mice; SEKI, $N = 3$ mice; CRCA, $N = 3$ mice; STRV, $N = 2$ mice. **d** Scatterplots of changes (log2) in CRCA liver mRNA gene expression data (Y-axis) with estimated enzyme protein concentrations (X-axis). Transcriptome data; CRCA, $N = 4$ mice; C57BL/6N, $N = 4$ mice. **e** Density plot of enzyme concentrations. High concentrations (gray zone), protein concentrations above the highest quartile of a published liver proteome. **f** Scatterplots of metabolites and Gly enzymes. MEWO, blue; LNCA, green; SEKI, orange; CRCA, red; STRV, purple. Dashed lines are produced by linear regression without the STRV point. Gly enzymes, enzymes that catalyze Gly directly, NIA niacin, B6 vitamin B6-related metabolites. r Pearson's correlation coefficient; P non-adjusted P-value (Pearson's correlation). **g** Schematic of changing hepatic C1 metabolism. Gly, Gly-related enzymes, and metabolites; SAM, SMA-related enzymes, and metabolites. Source data are provided as a Source Data file.

## Enzyme-linked immunosorbent assay (ELISA)

We collected mouse plasma using Ethylenediamine-N, N, N', N'-tetra-acetic Acid Dipotassium Salt Dihydrate (2 K) (FUJIFILM Wako, CAS RN:25102-12-9). Mouse plasma SAA (Serum Amyloid A), Albumin, IL-6 (Interleukin 6), and GDF15 (Growth and Differentiation Factor 15) were quantified in duplicate using commercial ELISA kits (Mouse Serum Amyloid A Quantikine ELISA kit (R&D systems, Cat# MSAA00; lot number, not recorded; dilution, NA), Mouse Albumin ELISA kit (Abcam, Cat# ab207620; lot number, not recorded; dilution, NA), Mouse IL-6 ELISA kit (Merk, Cat#RAB0308; lot number, not recorded; dilution, NA), Mouse/Rat GDF-15 Quatikine ELISA kit (R&D systems, Cat#MGD150; lot number, not recorded; dilution, NA)). The absorbance of each well at 450 nm was determined with a Multiskan FC Microplate Photometer (Thermo Fisher Scientific, MA, USA; Catalog number: 51119000). For plasma IL-6 quantification, undetectable values were regarded as 0 pg/mL ($n = 11$ samples), and values below the minimum detectable dose (MDD) of IL-6 (2 pg/ml) were included without adjustment ($n = 6$ samples).

## RNA extraction and microarray

Liver samples were collected from four CRCA and four C57BL/6N mice. Samples were immersed in RNAlater (Thermo Fisher Scientific) overnight at 4 °C and stored at −80 °C. According to the manufacturer's instructions, total RNA was isolated from samples using an RNAeasy Mini Kit (QIAGEN) with on-column RNase-free DNase I (Qiagen) treatment. Cy3-labeled cRNA was prepared from 100 ng total RNA using a Low Input Quick Amp Labeling Kit (Agilent) according to Agilent's standard protocol. Labeled cRNA was hybridized to a SurePrint G3 Mouse GE 8x60K Microarray using the Gene Expression Hybridization Kit (Agilent). Agilent Microarray Scanner was used to scan array. Images were quantified using Agilent Feature Extraction software. Quantified raw data were normalized by the quantile method using R with the limma package. Microarray data are deposited in the ArrayExpress database at EMBL-EBI under accession number E-MTAB-11771.

## Immunoblotting

Frozen mouse tissue samples were lysed in urea-based buffer (20 mM HEPES (pH 8.0), 9 M urea, 1 mM sodium orthovanadate, 2.5 mM sodium pyrophosphate, 1 mM β-glycerophosphate) and were sonicated using a sonicator (MICROSON XL-2000; Misonix, Farmingdale, NY) for -30 s on ice. Protein concentration was quantified with the Bradford assay (BIO-RAD, Hercules, CA, USA). 10 μg of total protein per sample were applied to 5-20% SuperSep Ace gradient gels (FUJIFILM Wako) and separated by SDS-PAGE. Samples were transferred to Immobilon PVDF membranes (Millipore, Burlington, MA, USA). After blocking with Tris-based saline with 0.1% Tween-20 and 5% w/v nonfat dry milk (Cell Signaling Technology MA, USA; Cat#9999), membranes were incubated overnight with Malonyl-Lysine [Mal-K] MultiMab Rabbit mAb mix (Cell Signaling Technology, Cat#1492S; RRID: AB_2687627; lot number, 3; dilution, 1:1000) or anti-Malonyllysine

Rabbit pAb (PTM BIO; Cat#PTM-901; lot number, #ZC0621105P0; dilution, 1:1000). The primary antibodies were detected with anti-Rabbit HRP-conjugated secondary antibody (Cell Signaling Technology, Cat#7074; RRID: AB_2099233; lot number, not specified; dilution, 1:3000). To detect GAPDH protein, we used anti-GAPDH antibody conjugated with HRP (FUJIFILM Wako, Cat#015-25473; RRID: AB_2665526; lot number, not specified; dilution, 1:10000). Signals were visualized using the Immobilon western detection system (Millipore), and chemiluminescent images were obtained with an ImageQuant LAS 4000 mini (GE Healthcare, Chicago, IL, USA).

## Transmission electron microscopy

For electron microscopy analysis, we used the service of Tokai Electron Microscopy (Nagoya, Japan). The contractor provided the following protocol. Liver samples were fixed with 2% paraformaldehyde and 2% glutaraldehyde in 0.1 M phosphate buffer (PB) pH 7.4 at 4 °C overnight. Next, samples were washed 3 times in 0.1 M PB and incubated with 0.1 M PB for 30 min, and were then post-fixed with 2% osmium tetroxide in 0.1 M PB at 4 °C for 2 h. After post-fixation, samples were dehydrated through the following solutions: 50% ethanol for 20 min at 4 °C; 70% ethanol for 20 min at 4 °C; 90% ethanol for 20 min at room temperature (RT); 4 changes of absolute ethanol, 20 min each at RT. After dehydration, samples were infiltrated with propylene oxide 2 × 30 min and were put into 70:30 mixtures of PO and resin (Quetol-812; Nisshin EM Co. Tokyo, Japan) for 1 h and left to volatilize PO overnight. Samples were then immersed in fresh 100% resin and polymerized at 60 °C for 48 h. Samples embedded in resin were cut to 70 nm with an ultramicrotome equipped with a diamond knife (Ultracut UCT; Leica). Sections were stained with 2% uranyl acetate at room temperature for 15 min and were washed with distilled water followed by secondary staining with a lead stain solution (Merck) at room temperature for 3 min. Samples were observed with a transmission electron microscope (JEM-1400Plus; JEOL Ltd., Tokyo, Japan) at an acceleration voltage of 100 kV. Digital images were captured with a CCD camera (EM-14830RUBY2; JEOL Ltd). We adjusted the brightness and contrast of electron microscopic images presented in Fig. 3a, b with Affinity Photo (version, 1.10.6.1665; Serif, Nottingham, UK).

## Metabolomic analysis

Metabolomic analysis was performed following the protocol described in our published study[19]. Specifically, -50 mg each of cryopreserved muscle and liver samples was accurately weighed, recorded, and thoroughly homogenized with a cell disrupter (Shake Master NEO; Bio Medical Science, Tokyo, Japan) at 1500 rpm for 5 min in 500 μL of methanol-based extract solution containing 20 μM each of methionine sulfone, MES (2-(N-morpholino)-ethanesulfonic acid), and CSA (D-camphor-10-sulfonic acid) as internal standards. Blood samples were simply mixed with 500 μL of methanol-based extract solution containing 20 μM each of methionine sulfone, MES, and CSA without homogenization. The processed sample solution was mixed with

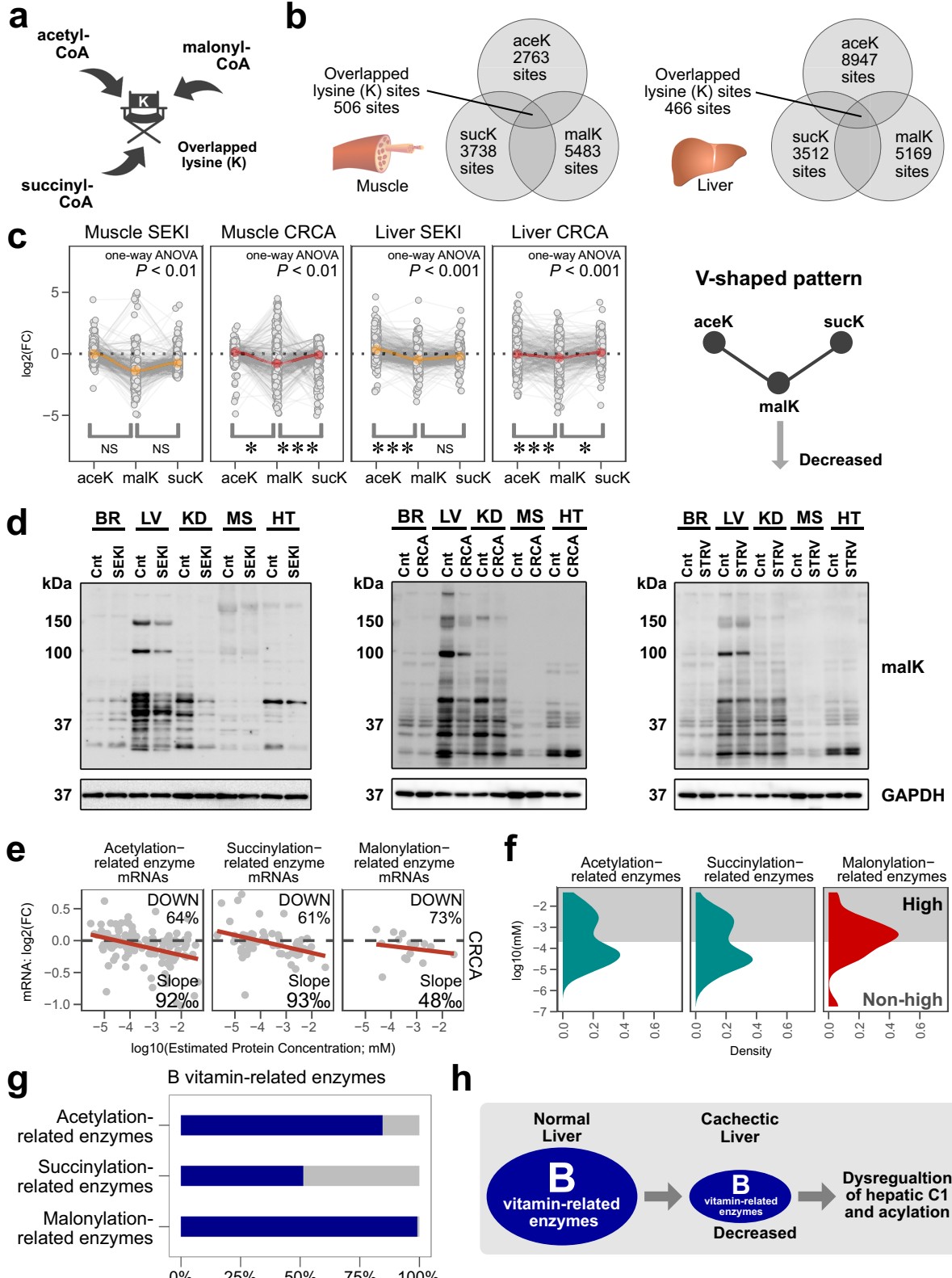

500 μL of chloroform and 200 μL of ultrapure water and centrifuged at 4600 g for 15 min at 4 °C. 300 μL of the aqueous solution was taken from the upper water-MeOH layer and run through a centrifugal 5000-Da cut-off ultrafiltration filter (Human Metabolome Technologies, Tsuruoka, Japan) at 9000 g and 4 °C overnight. The centrifugally concentrated filtrate was dissolved in 50 μL of ultrapure water containing 200 μM each of 3-aminopyrrolidine and trimesate and

immediately subjected to Capillary Electrophoresis-Mass Spectrometry (CE-MS).

We quantified concentrations of charged metabolites in the pre-processed samples using Capillary Electrophoresis-Time-of-Flight Mass Spectrometry (CE-TOFMS) equipment (Agilent Technologies, CA, USA) according to the methods established by the authors[67,68]. Cationic compounds were analyzed employing a fused silica capillary

**Fig. 7 | Systemic decline of protein lysine malonylation in SEKI and CRCA. a** Schematic diagram of competition among three reactive acyl-CoA molecules at a single lysine (K) site. **b** Venn diagram showing numbers of overlapping lysine (K) residues for acetylation (aceK), malonylation (malK), and succinylation (sucK) in SEKI and CRCA muscle and liver data. **c** Fold changes in proteomic data for lysine sites shared by acetylation (aceK), malonylation (malK), and succinylation (sucK). *$P < 0.05$, ***$P < 0.001$, NS, $P > 0.05$. by Dunnett's 2-sided test with malK as a single control after one-way ANOVA. Orange and red circles, median values. Note that only proteins with an adjusted FC >0.03 and <32 were presented. SEKI, $N = 3$ mice; CRCA, $N = 3$ mice. Analyzed overlapped lysine residues; muscle SEKI, 456 residues; muscle CRCA, 437 residues; liver SEKI, 404 residues; liver CRCA, 369 residues. *P*-values (one-way ANOVA); muscle SEKI, 0.006388; muscle CRCA, 0.00438; liver SEKI, $2.2 \times 10^{-16}$; liver CRCA, $6.32 \times 10^{-6}$. Adjusted *P*-values (Dunnett's 2-sided test); muscle SEKI, malK vs aceK, 0.0553; muscle SEKI, malK vs sucK, 0.549; muscle CRCA, malK vs aceK, 0,0975; muscle CRCA, malK vs sucK, 0.00204; liver SEKI, malK vs aceK, $1.0 \times 10^{-10}$; liver SEKI, malK vs sucK, 0.499; liver CRCA, malK vs aceK,

$2.03 \times 10^{-6}$; liver CRCA, malK vs sucK, 0.0167. **d** Western blotting analysis using anti-malonyl-lysine (malK) antibody mixture. SEKI, $N = 1$ mouse; CRCA, $N = 1$ mouse; STRV, $N = 1$ mouse. BR, brain; LV, liver; KD, kidney; MS, muscle; HT, heart; Cnt, control mice; GAPDH, glyceraldehyde 3-phosphate dehydrogenase; kDa, kilodalton. GAPDH is a loading control. **e** Decline of mRNAs coding for acetylation, succinylation, and malonylation-related enzymes in CRCA liver. Note that only mRNAs with estimated protein concentrations exceeding $1.0 \times 10^{-6}$ mM are presented. Transcriptome data; CRCA, $N = 4$ mice; C57BL/6N, $N = 4$ mice. The calculation of the protein concentration estimates is described in the Methods section (Methods, Estimation of protein concentration). **f** Density plots of public proteome data. High concentrations (gray zone), protein concentrations above the highest quartile of a published liver proteome. **g** Percentage of B vitamin-related enzymes to total amounts of specified enzyme groups in liver (dark blue). **h** Schematic of decreased liver B vitamin-related enzymes in cancer cachexia. Source data are provided as a Source Data file.

(inner diameter, 50 µm; length, 1 m) filled with 1 M formic acid as the leading electrolyte and 50% (v/v) methanol-water mixture containing 0.1 µM hexakis (1H,1H,2H-perfluoroethoxy) phosphazene as the sheath liquid[69]. The sheath liquid was delivered into the capillary at a constant rate of 10 µL/min. Electrospray ionization time-of-flight mass spectrometry (ESI-TOFMS) was performed under positive-on mode conditions, with the capillary voltage set at 4000 V. Acquired spectrums were automatically recalibrated based on the masses of reference standards ([[13C] isotopic ion of a protonated methanol dimer (2 MeOH +H)]+, m/z 66.0631) and ([hexakis (1H,1H,2H-perfluoroethoxy) phosphazene +H]+, m/z 622.0290). Metabolites were identified by m/z values and relative migration times normalized to the migration time of 3-aminopyrrolidine. Metabolite concentrations were calculated by comparing peak areas against a calibration curve constructed using internal standardization techniques with methionine sulfone. Unless specified otherwise, our CE-MS analytical procedures for cationic metabolites were the same as previously described[68].

Anionic metabolites were analyzed employing a capillary coated with cationic polymer (COSMO(+) Capillary; inner diameter, 50 µm; length 105 cm; Nacalai Tesque, Kyoto, Japan) filled with 50 mM ammonium acetate buffer (pH 8.5) as the leading electrolyte and 50% (v/v) methanol-5 mM ammonium acetate mixture containing 0.1 µM hexakis (2,2-difluoroethoxy) phosphazene as the sheath liquid. The sheath liquid for anionic metabolites was delivered at 10 µL/min. ESI-TOFMS was performed under negative-ion mode conditions, with the capillary voltage set at 3500 V. Trimesate and CAS were utilized as reference and internal standards to analyze anionic metabolites. Unless specified otherwise, our analytical procedures for anionic metabolites using CE-MS remained consistent with the method previously described[67].

We processed the raw data from CE-TOFMS equipment, including binning data into 0.02 m/z slices and peak picking, using our in-house software tool, MasterHands (version 2.17.0.10.). Subsequently, our software created data matrices through an alignment process based on adjusted migration times, then assigned metabolite names to aligned peaks by matching m/z and adjusted migration times with our standards library and calculated relative peak areas by dividing by the peak of the internal standard. Metabolite concentrations were finally determined based on the relative peak area between the sample and the standard mixture.

## Sample preparation for LC-MS analysis

For tandem mass tag (TMT) labeling (Thermo Fisher Scientific), samples were pretreated basically using the phase transfer surfactant-aided trypsin digestion method[70]. Specifically, frozen mouse tissue samples were lysed in Tris-based buffer with phase-transfer surfactants (50 mM Tris-HCl (pH 7.5), 150 mM NaCl, 1 mM EDTA, 1 mM PMSF, 1 mM

DTT, Protease Inhibitor, PhosSTOP, 12 mM SDC (sodium deoxycholate), 12 mM SLS (sodium lauroyl sarcosinate)) and were sonicated using a sonicator (MICROSON XL-2000; Misonix, Farmingdale, NY) for ~30 s on ice. The lysate was centrifuged, and the separated supernatant was centrifuged again to collect the supernatant for the next step. Protein concentration was quantified with the Bradford assay (BIO-RAD). To digest proteins with Trypsin Gold (Cat# V5280; Promega, Madison, WI, USA) and Lysyl endopeptidase (Lys-C; Cat# 125-05061, FUJIFILM Wako), the supernatant with 100 µg protein was denatured at 95 °C for 5 min. After sonication using a Bioruptor (BM Equipment, Tokyo, Japan) for 10 min on ice, the solution was centrifuged, and the supernatant was collected. DTT was added to the supernatant (final concentration 10 mM; Cat# 048-29224, FUJIFILM Wako) and incubated for 30 min at room temperature. After reduction with DTT, iodoacetamide was added to the samples (final concentration 40 mM; Cat# I6125, Sigma), incubated for 1 h in the dark at room temperature, and then 5 volumes of 50 mM Tris-HCl (pH 8.5) were added. Samples were incubated with Lys-C for 3 h at 37 °C before incubation with Trypsin Gold for 16 h at 37 °C. Equal volumes of ethyl acetate were then added to samples followed by trifluoroacetic acid (TFA, final concentration 0.1%) to remove SDC and SLS. After desalting with C18 (GL-Tip SDB; Cat#7820-11200, GL Science, Tokyo, Japan), peptide concentrations were determined. Samples were divided into aliquots containing 25 µg of peptides, and evaporated in a Savant SpeedVac. Samples were then dissolved in 25 µL of 50 mM triethylammonium bicarbonate solution. Following centrifugation, 8.2 µL of TMT reagent were added to the 20-µL samples at room temperature, mixed well, and allowed to react for 1 h at room temperature followed by addition of hydroxylamine to stop the reaction. After evaporating the acetonitrile in the TMT reaction solution, the solid phase was dissolved in a urea-based buffer solution (2 M urea and 1% TFA). The solution was then divided into seven fractions by GL-Tip SDB-SCX (Cat # 7510-11202, GL Sciences)[71].

## Sample preparation for post-translational modification (PTM) analyses

For PTM analyses, we used PTMScan Acetyl-Lysine [Ac-K] Kit (Cat# 13416, Cell Signaling Technology; lot number, 6; dilution, NA), PTMScan Succinyl-Lysine [Succ-K] Kit (Cat# 13764, Cell Signaling Technology; lot number, 2; dilution, NA), and PTMScan Malonyl-Lysine [Mal-K] Kit (Cat# 93872, Cell Signaling Technology; lot number, 1; dilution, NA) and followed the kit's protocols. Briefly, samples were lysed in a urea-based buffer (20 mM HEPES (pH 8.0), 9 M urea, 1 mM sodium orthovanadate, 2.5 mM sodium pyrophosphate, 1 mM β-glycerophosphate) and were sonicated using a sonicator for ~30 s on ice. After centrifuging the lysate, the supernatant was collected for further preparation steps instructed by the manufacturer's protocol.

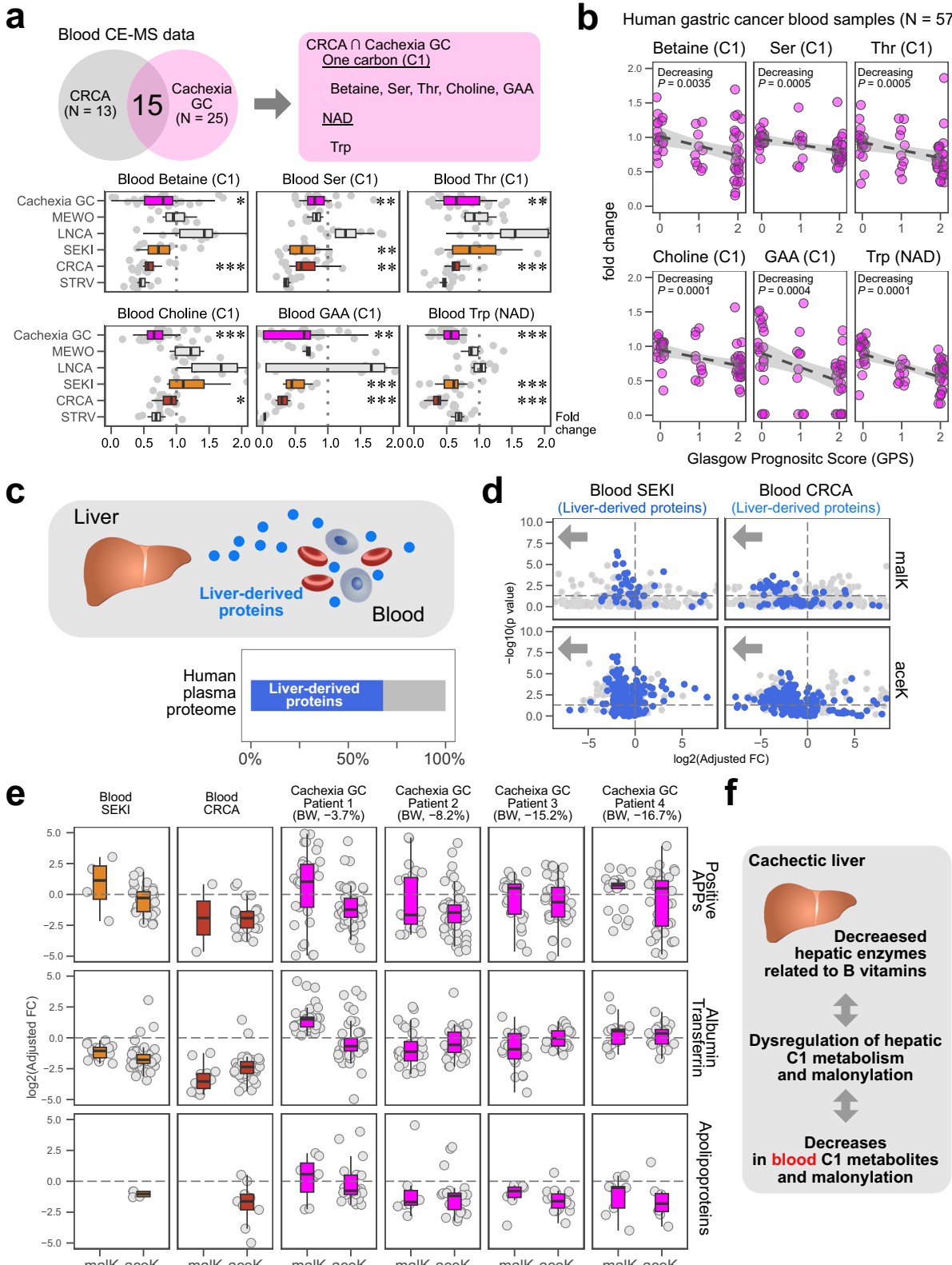

## LC-MS analysis

Purified samples were analyzed by LC-MS using a Q Exactive mass spectrometer (Thermo Fisher Scientific) coupled to an Ulti-Mate3000 RSLCnano LC system via a nanoelectrospray ion source. Peptides were separated on a 150-mm nano HPLC capillary column (inner diameter 75 μm, Nikkyo Technos Co., Tokyo, Japan) and loaded into a reversed-phase chromatography column with buffer A (2% acetonitrile with 0.1% formic acid) and eluted with a 100-min linear gradient from 5%-40% buffer B (90% acetonitrile with 0.1% formic acid) at an estimated flow rate of 300 nL/min. A precursor ion scan was conducted using a 400-1600 mass to charge ration (m/z) before MS2 analysis.

**Fig. 8 | Decreased levels of C1-related metabolites and protein malonylation in blood samples from severe cachexia mouse models and advanced gastric cancer cachexia patients. a** C1- and NAD- related metabolites commonly altered in blood samples of CRCA mice and clinical cachexia patients (Welch's 2-sided $t$-test, $P$-value < 0.05). These graphs show levels of betaine, serine (Ser), threonine (Thr), choline, guanidinoacetate (GAA), and tryptophan (Trp) with respect to controls. GC, gastric cancer. Cachexia GC, $N = 25$ patients; MEWO, $N = 6$ mice; LNCA, $N = 10$ mice; SEKI, $N = 8$ mice; CRCA, $N = 13$ mice; STRV, $N = 8$ mice. Welch's 2-sided $t$-test presented only for cachexia GC, SEKI, and CRCA; *$P < 0.05$, **$P < 0.01$, ***$P < 0.001$. Non-adjusted $P$-values (Welch's 2-sided $t$-test); Cachexia GC, betaine (0.0145), Ser (0.0135), Thr (0.00690), choline ($5.32 \times 10^{-5}$), GAA (0.000254), Trp ($1.13 \times 10^{-10}$); SEKI, betaine (0.118), Ser (0.00492), Thr (0.749), choline (0.184), GAA ($2.51 \times 10^{-5}$), Trp ($5.99 \times 10^{-5}$); CRCA, betaine ($7.65 \times 10^{-6}$), Ser (0.00122), Thr (0.000757), choline (0.0324), GAA ($1.1 \times 10^{-5}$), Trp ($6.88 \times 10^{-17}$); Boxplot style, the middle line (median), the upper hinge (Q3, the third quantile), the lower hinge (Q1, the first quantile), the upper whisker ($1.5 \times$ (Q3 − Q1)), the lower whisker ($1.5 \times$ (Q3 − Q1)). **b** Integrated fold changes of blood metabolites from early and advanced gastric cancer patients with respect to those from reference samples. GPS 0, $N = 20$ patients; GPS 1, $N = 10$; patients; GPS 2, $N = 27$ patients. Dashed lines, linear model; gray area, 95% confidence interval. $P$-values by Jonckheere's trend test (one-sided) for GPS order. **c**, Percentage of liver-derived proteins in human plasma proteome data weighted by protein abundance (3461 human plasma proteins). **d** Volcano plots of malonyl-K (malK) and acetyl-K (aceK) data for blood proteins in SEKI and CRCA mice. Blue points, liver-derived proteins; gray points, other proteins. SEKI, $N = 3$ mice; CRCA, $N = 3$ mice. Adjusted FC, fold changes adjusted by total protein fold changes. Horizontal dashed lines, $P = 0.05$ (Welch's 2-sided $t$-test). **e** Box plots with horizontally jittered points for fold-changes of malK and aceK peptides from liver-derived blood proteins. Change values were calculated from SEKI mouse datasets ($N = 3$ mice) with control nude mice ($N = 3$ mice), CRCA mice ($N = 3$ mice) with control C57BL/6N mice ($N = 3$ mice), and gastric cancer cachexia patients ($N = 4$ patients) with control early gastric cancer patients without cachexia ($N = 6$ patients). Percentage values for BW (body weight) indicate each patient's weight changes between blood collection timepoints and -6 months before. Adjusted FC, fold changes adjusted by fold changes of total protein levels. APPs, acute-phase proteins. Note that only proteins with an adjusted FC >0.03 and <32 and a $P$-value (Welch's 2-sided $t$-test) <0.05 are presented. Boxplot style, the middle line (median), the upper hinge (Q3, the third quantile), the lower hinge (Q1, the first quantile), the upper whisker ($1.5 \times$ (Q3 − Q1)), the lower whisker ($1.5 \times$ (Q3 − Q1)). **f** Schematic of cachexia-associated metabolic changes in the liver and blood. Source data are provided as a Source Data file.

## MS data analysis

Raw MS data from TMT labeling proteomics and PTM analyses were processed using Proteome Discoverer 2.2 (Thermo Fisher Scientific), followed by the MASCOT search engine, version 2.4 (Matrix Science, Boston, MA, USA). Peptides and proteins were searched against the Mouse UniProt FASTA database (February 17, 2020), containing 17,026 entries with a precursor mass tolerance of 10 ppm and a fragment ion mass tolerance of 0.02 Da. The search for TMT labeling data included fixed modifications of carbamidomethyl cysteine and variable modifications of methionine oxidation, TMT6plex (N-term), and TMT6plex(K). One missed cleavage by trypsin was allowed. The false discovery rate (FDR) was set to 1% for peptide and protein identification. The ratio of the median intensity of the sample to the median intensity of the reference sample was taken as the ratio of relative protein abundance. Data were normalized by the median ratio normalization method provided by MASCOT. The search for PTM data included fixed modifications of carbamidomethyl cysteine and variable modifications of oxidation and lysine acetylation, lysine malonylation, or lysine succinylation. Four missed cleavages by trypsin were allowed. Search results of PTM technical triplicate data by MASCOT were imported into Skyline (version 20.1, MacCoss Lab Software, University of Washington) to create MS/MS spectral libraries used for MS1 filtering[72,73]. PTM data were also normalized by the median ratio normalization method using the free software R (version 4.1.3 (2022-03-10), R Foundation for Statistical Computing, Vienna, Austria) with RStudio IDE (version 2022.07.01 + 554; Posit, PBC).

## Data analysis and visualization

No new algorithms were developed for this study. Data analysis and visualization were performed using R (version 4.1.3 (2022-03-10), R Foundation for Statistical Computing, Vienna, Austria) [https://www.r-projectorg/] with RStudio IDE (version 2022.07.01 + 554; Posit, PBC). Single-channel mRNA microarray data of mouse livers were normalized by the quantile method using the limma package (version, 3.50.3). Metabolites for which all samples in a data set were missing were removed, and all other missing values were replaced by one-half of the minimum concentration of all variables in the data set[74]. As for proteome data, protein variables containing missing values were removed. Unless otherwise stated, points outside the X and Y scale ranges were analyzed, but simply not visualized. To create graphics, charts, and schematics, we used PowerPoint (version, 2307; Microsoft, Redmond, WA, USA), Affinity Photo (version, 1.10.6.1665; Serif), PyMOL (version 2.5.2; Schrodinger, New York, NY, USA), ggplot2 (version, 3.4.2) through tidyverse (version, 2.0.0), ggthemes (version, 4.2.4), viridis (version, 0.6.2), patchwork (version, 1.1.2), and survminer (version, 0.4.9). For empirical Bayes moderated $t$-statistics test, we used the eBays function from the limma package (version, 3.50.3). For Welch's 2-sided $t$-test, we used t.test function in the base R stats package (version, 4.1.3). For Steel's 1-sided test and Dunnett's 2-sided test, we used steelsKSampleTest function (alternative; less or greater) from the PMCMRplus (version, 1.9.7) and the glht function from the multcomp (version, 1.4-23) package, respectively. For principal component analysis (PCA), the prcomp function in the base R stats package (version, 4.1.3) was used. Variables were scaled. For linear support vector machines (SVMs), we used the svm function in the e1071 package (version, 1.7.13), with the cost and gamma parameters set to default values. In Fig. S5f, maximal margins were the shortest Euclidean distance between the two margins on PCA score plots. For Random Forests, the randomForest function (ntrees = 500, importance = TRUE) from the the randomForest package (version, 4.7-1.1) in R was used. We used type = 1 and scale = TRUE in the importance function call to calculate mean decreasing accuracy. Two thousand Random Forest trials without setting the initial seed for random generator were performed to calculate the 95% confidence interval of MDA (mean decreasing accuracy) for each metabolite. Correlation matrix (Fig. S4b) was created by the ggcorrplot function from the ggcorrplot package (version, 0.1.4). Pearson's correlation coefficients were calculated using cor.test function provided by base R (version, 4.1.3) (Figs. 4b–d, 6f, S3e, S4b, S6b, S7f, and S7h). Linear regression slopes calculated on log-log plots were regarded as slope angles: Figs. 5b, 6d, 7e, S7e–g, and h. Generalized variance-inflation factors were calculated by vif function of car package (version, 3.1.2): Fig. 5d. Kaplan-Meier survival analyses were performed using the survfit function of the survival package (version, 3.2-13): Fig. S1a, S2a, S3a, b. Jonckheere-Terpstra test was performed using the jonckheere.test function (nperm = 1000) from the clinfun package (version, 1.1.1): Fig. 8b. Univariate and multivariate ROC (receive-operating characteristic) analyses were performed using the ROC function from the pROC package (version, 1.18.2) and glm function (Fig. S10). Mosaic plots were created using the mosaicplot function provided by base R (version, 4.1.3) (Fig. S11a, b).

## Calculation of fold changes (FC) and sample numbers

This study focused on fold change values between experimental and control groups. The sample size used to calculate fold changes is as follows (experimental group size in the numerator/control group size in the denominator). Mouse body composition data; female MEWO, 17/17; male MEWO, 14/14; female LNCA, 11/11; male LNCA, 11/

11; female SEKI, 6/5; male SEKI, 6/7; female CRCA, 10/10; male CRCA, 6/6; female STRV, 6/6; male STRV, 6/6. Mouse muscle real time PCR data; MEWO, 5/5; LNCA, 5/5; SEKI, 5/4; CRCA, 5/5; STRV, 6/6. Mouse muscle metabolome (CE-MS); MEWO, 6/6; LNCA, 11/14; SEKI, 8/10; CRCA, 13/25; STRV, 8/8; FK866, 11/12; 5FU, 5/5. Mouse liver metabolome (CE-MS); MEWO, 6/6; LNCA, 11/14; SEKI, 11/12; CRCA, 13/25; STRV, 8/8; FK866, 11/12; 5FU, 5/5. Mouse liver metabolome (CE-MS) for Fig. S3e; CRCA under 10 weeks of age, 8/7; CRCA over 12 weeks of age, 12/12. Mouse plasma metabolome (CE-MS); MEWO, 6/6; LNCA, 10/14; SEKI, 8/10; CRCA, 13/21; STRV, 8/5. In Fig. 8a, human blood metabolome (CE-MS); 25/14, cachexia advanced gastric cancer patients/non-cachexia early gastric cancer patients as a reference control group. In Fig. 8b, human blood metabolome (CE-MS); 57/14, gastric cancer patients/non-cachexia early gastric patients as a reference control group. Mouse muscle proteome (LC-MS; TMT labeling); MEWO, 3/2; LNCA, 3/3; SEKI, 3/3; CRCA, 3/3; STRV, 2/2. Mouse liver proteome (LC-MS; TMT labeling) except Fig. S8; MEWO, 3/2; LNCA, 3/3; SEKI, 3/3; CRCA, 3/3; STRV, 2/2. Mouse liver proteome (LC-MS; TMT labeling) only for Fig. S8; female SEKI, 3/2; male SEKI, 3/2; female CRCA, 3/2; male CRCA, 3/2. Mouse blood proteome (LC-MS; TMT labeling); SEKI, 3/3; CRCA, 3/3. Human blood proteome (LC-MS, TMT labeling); 6/6, cachexia advanced gastric cancer patients/non-cachexia early gastric cancer patients as a reference group. Mouse muscle proteome (LC-MS; PTM scan); SEKI, 3/3; CRCA, 3/3. Mouse liver proteome (LC-MS; PTM scan); SEKI, 3/3; CRCA, 3/3. Mouse blood proteome (LC-MS; PTM scan); SEKI, 3/3; CRCA, 3/3. Human blood proteome (LC-MS, PTM scan); 4/6, cachexia advanced gastric cancer patients/non-cachexia early gastric cancer patients as a reference group. In Fig. 8d, e, individual fold change values of PTM data were adjusted by individual total protein changes obtained by TMT labeling quantitative proteomic analysis.

### Enrichment analysis

We performed enrichment analysis using aGOtool[75]. In Fig. S5b, UniProt keyword-based analyses were conducted. Proteins with PC1 loading >0.6 in the liver proteome and proteins with PC2 loading <−0.6 in the skeletal muscle proteome were used as the foreground proteome: PC loading data are provided in the Source Data file for Fig. S5b. Multiple testing was corrected with the Benjamini-Hochberg method. S value is the same as the pi-value[76] (personal communication with Dr. David Lyon, the aGOtool developer).

### Annotation-based data retrieval and reaggregation

To generate "biological objects" in this study, we aggregated data according to search terms and formulas in Table S5 using R with the tidyverse (version, 2.0.0) and stringr (version, 1.4.1) packages. The simple addition of concentrations produced the aggregated values. For proteomic data, we searched the UniProt ID, Catalytic activity, EC number, Nucleotide binding, Keyword ID, and ChEBI IDs columns of UniProt Knowledgebase downloaded in 2022 [https://www.uniprot.org/]. The combination of search terms was based on educated guesses. When performing enzyme-specific analyses, we used a subset filtered by the "All enzymes" search formula described in Table S5. In Fig. 3d, liver-specific enzymes refer to enzymes that are mouse homologs of human liver-specific genes based on the Human Proteome Atlas[77] and are not included in Ribosome proteins, Endoplasmic proteins, Golgi apparatus, or Immunity including APPs. In Fig. 8c, the percentage of liver-derived proteins in human plasma was estimated by integrating the PAXdb human plasma data with published data from Franko et al. [78]. We removed possible contaminating proteins according to criteria of Franko et al. In Fig. 8e, liver-derived apolipoproteins were selected by Franko's list. In Fig. S11, we considered common proteins to be those whose protein parts of UniProt Entry Name are shared between mouse and human proteomes. The same process determined liver-derived proteins as in Fig. 8e.

### Reference concentrations of mouse liver metabolites

In Figs. 2d, 4b–e, metabolite concentrations in mouse liver are mean values of liver CE-MS metabolomic data obtained from control C57BL/6 strain mice ($N = 59$ mice). The unit of metabolite concentration by CE-MS was presented in moles per unit tissue weight. Assuming a tissue-specific gravity of $1.0 \, g/cm^3$, we transformed the unit into a molar concentration (BNID115456)[60]. For biotin, folate, and vitamin B12 (B12), values collected from the literature were used. For folate and B12, values from mouse liver were used[79,80], and for biotin, values from rat liver were adopted. For vitamin unit conversions, ignoring the presence of multiple derivatives, we used the following molecular masses: biotin, 244.31; folate, 441.4; B12, 1355.388. In addition, for the unit conversion of biotin based on rat liver data, we assumed that the total protein concentration in rat liver was 310 mg per g of tissue, and the tissue-specific gravity was $1.0 \, g/cm^3$ (BNID113242 and BNID115456)[60].

### Details of estimation of B vitamin concentrations from CE-MS data or food datasets

For simplicity, the following molecular weights were used in molar calculations: B1, 265.355; B2, 376.36; NIA, 123.111; B6, 169.18; PA, 219.23; Biotin, 244.31; Folate, 441.4; B12, 1355.388. For Figs. 2b, 4d, 4e, 6f, S3e S4a, and S6b, concentrations of mouse B vitamins were calculated by adding the concentrations of individual B-vitamin-related metabolites quantified by CE-MS, as mentioned in the Annotation-based Data Retrieval and Reaggregation section and Table S5. In Figs. 4d, e, regarding *Saccharomyces cerevisiae*, we used food nutrition profile data (Yeast, baker's yeast, compressed, Item No.17082, Standard Tables of Food Composition in Japan, 2015; Ministry of Education, Culture, Sports, Science and Technology, Japan, https://www.mext.go.jp/). B vitamin concentrations of *Bos taurus* represent the mean values of liver, heart, kidney, small intestine, and rump food composition values (Item No.11092, No.11091, No.11093, No.11098, and No.11028; Standard Tables of Food Composition in Japan, 2015). For *Arabidopsis thaliana*, a member of the family Brassicaceae, average values of three Brassicaceae were extrapolated (*Capsella bursa-pastoris*, Item No.06200; *Brassica rapa*, Item No.06201; *Brassica napus*, Item No.06203, Standard Tables of Food Composition in Japan, 2015).

### Estimation of protein concentrations

The average of label-free quantification intensity (LFQ) values from biological quadruplicates (LFQ intensity Liver_R1-R4, 1-s2.0-S1550413114004999-mmc2.xlsx[41]) was adopted to estimate molar concentrations of mouse hepatic proteins in the following figures: Figs. 4a, b, 5a–d, 6a–f, 7e–g, S6b, S6c, S7b, S7c, S7e–h, S8a, b, S9c, and S9d. The PAXdb datasets were used in the following figures: Figs. 4c–e, and 8c. We used mouse liver PAXdb data in Figs. 4c–e, and human plasma data in Fig. 8c. All PAXdb datasets were downloaded in 2022. To estimate concentrations of individual proteins, we assumed the total protein concentration to be 5 mM in this study[81].

### Linking from metabolome to proteome

When linking a metabolite to related proteins in Fig. S6c, we used the IDs of Chemical Entities of Biological Interest (ChEBI) retrieved from the UniProt database. We manually linked the IDs of ChEBI preferred in the UniProt to our CE-MS data metabolite names. The ChEBI and KEGG lookup table is also provided in the Source Data file for Fig. S6C.

### Statistics and reproducibility

The empirical Bayes moderated *t*-statistics test was used in Fig. 1c. In Fig. 1c, six experimental and six control group samples were randomly selected from pooled experimental and control datasets except for MEWO data, followed by the empirical Bayes test. Welch's 2-sided *t*-test results were used in the following table and figure panels: Table S2, Figs. 3d, 4a, 8a, 8d, S5c, S5d, S7b, S9c, and Table S2. The log-rank test

was used for Kaplan-Meier survival analyses (Fig. S1a, S2a, S3a, and S3b). Pearson's correlation was used in the following figure panels: Figs. 4b–e, 6f, S3e, S4b, S6b, S7f, and S7h. Multiple regression analyses of logarithmic and dummy variables were performed: Fig. 5d. Steel's 1-sided test was used in Table S3. Dunnett's 2-sided test was used in Fig. 2b (Table S4) and 7c. Jonckheere trend one-sided test with 10,000 permutations was used in Fig. 8b. Fisher's exact test was used in Fig. S11a, S11b, and Table S6. A $p$-value < 0.05 was considered significant.

For mouse experiments, small pilot experiments were conducted before main experiments to optimize experimental conditions and evaluate general mouse conditions. The mouse phenotypes of the optimized pilot experiments were confirmed in the following main experiments.

Omics analyses in this study were performed with independent biological replicates. Regarding the biological replicates, their size was based on our previous studies[19,82,83] and preliminary experimentation, and their details are described in the Methods section; Calculation of fold changes (FC) and sample numbers. The number of replicates is also stated in the figure legends. All the omics analyses with biological replicates and experimental controls produced acceptable and analyzable data in a single run. In addition, we validated each omics data against different mouse models or species data and literature-based knowledge. Therefore, we did not repeat omics experiments with some exceptions described below.

We obtained additional SEKI and CRCA liver proteome data for sex-based analysis with new independent biological samples. The repeated results are roughly the same as the first time (Fig. 5a and Fig. S8). Regarding SEKI and CRCA metabolomics analyses, we performed small-scale experiments using stable metabolic isotopes; four times for the SEKI liver metabolome, twice for the SEKI muscle metabolome, four times for the CRCA liver metabolome, and four times for the CRCA muscle metabolome. The repeated omics runs yielded results broadly similar to the first run. This manuscript presents the first SEKI and CRCA metabolome data with the largest number of biological replicates.

The B-vitamin cocktail administration experiment using SEKI model was performed only once because the first experiment result was not significant (Fig. S3a; Cnt, $N = 12$ female mice; B-vitamin cocktail (Bvc), $N = 12$ female mice; Log rank test, $P = 0.65$). The glycine administration experiment was performed twice, and the data were combined. The first experiment result was significant (Cnt, $N = 8$ female mice; Gly, $N = 8$ female mice; Log rank test, $P = 0.023$), but not the second experiment result (Cnt, $N = 16$ female mice; Gly, $N = 15$ female mice; Log rank test, $P = 0.29$). Fig. S3b shows the combined results of the first and second experiments (Cnt, $N = 24$ female mice; Gly, $N = 23$ female mice; Log rank test, $P = 0.021$).

Analysis of electron microscopic images (Fig. 3a–c) was intended to validate the enrichment analysis of hepatic proteomics data (Fig. S5b). We quantified electron microscopic images using ImageJ (1.54d)[84]. Cross-sectional areas of mitochondria and fat droplets were determined with ImageJ's freehand selection and oval selection modes, respectively. We used a $5.6 \times 4.2$ µm field of view to quantify mitochondrial cross-sectional area and expanded endoplasmic reticulum (Control, 168 fields (BALB/c nu/nu, $N = 2$ mice; C57BL/6, $N = 3$ mice; mitochondrial cross sectional area, $N = 1296$); MEWO, 29 fields (MEWO, $N = 1$ mouse; mitochondrial cross sectional area, $N = 253$); LNCA, 35 fields (LNCA, $N = 1$ mouse; mitochondrial cross sectional area, $N = 273$); SEKI, 38 fields (SEKI, $N = 1$ mouse; mitochondrial cross sectional area, $N = 171$); CRCA, 30 fields (CRCA, $N = 1$ mouse; mitochondrial cross sectional area, $N = 153$); STRV, 35 fields (STRV, $N = 1$ mouse; mitochondrial cross sectional area, $N = 124$)), and a $27.8 \times 20.9$ µm field of view to quantify fat droplet area (Control, 63 fields (BALB/c nu/nu, $N = 2$ mice; C57BL/6, $N = 3$ mice); MEWO, 13 fields (MEWO, $N = 1$ mouse); LNCA, 13 fields (LNCA, $N = 1$ mouse); SEKI, 13 fields (SEKI, $N = 1$ mouse); CRCA, 12 fields (CRCA, $N = 1$ mouse); STRV,

13 fields (STRV, $N = 1$ mouse)). The electron microscopic data presented in this study suggested that ultrastructural changes in the liver are associated with the severity of cancer cachexia-related phenotypes. These electron microscopic findings also support the liver proteomics changes (Fig. 3d; upregulation of ribosome and ER-related proteins). Note that we did not analyze the microscopic quantification data statistically due to the lack of independent biological replicants, except in the control group (Fig. 3c; Control, $N = 5$ mice; MEWO, $N = 1$ mouse; LNCA, $N = 1$ mouse; SEKI, $N = 1$ mouse; CRCA, $N = 1$ mouse; STRV, $N = 1$ mouse).

To further confirm the results of the PTM proteomics analyses (Fig. 7b), we planned western blotting analyses using malonyl-Lysine [Mal-K] MultiMab Rabbit mAb mix (Cell Signaling Technology, Cat#1492 S; RRID: AB_2687627; lot number, 3; dilution, 1:1000). As a pilot experiment, we initially analyzed SEKI and CRCA liver samples and observed downregulation of protein malonylation in the cachectic liver (Source Data file for Fig. 7c; BALB/c nu/nu, $N = 4$ mice; SEKI, $N = 4$ mice; C57BL/6N, $N = 4$ mice; CRCA, $N = 4$ mice). Next, we conducted another western blotting analysis using a different antibody, anti-Malonyllysine Rabbit pAb (PTM BIO, Cat#PTM-901; lot number, #ZC0621105P0; dilution, 1:1000) and obtained roughly similar western blotting results, re-confirming downregulation of protein malonylation in the cachectic liver (Source Data file for Fig. 7c; BALB/c nu/nu, $N = 4$ mice; SEKI, $N = 4$ mice; C57BL/6N, $N = 4$ mice; CRCA, $N = 4$ mice). Based on these validation experiments, we conducted the Fig. 7c's analyses (BALB/c nu/nu, $N = 1$ mouse; SEKI, $N = 1$ mouse; C57BL/6N, $N = 1$ mouse; CRCA, $N = 1$ mouse; C57BL/6N, $N = 1$ mouse; STRV, $N = 1$ mouse).

## Reporting summary
Further information on research design is available in the Nature Portfolio Reporting Summary linked to this article.

## Data availability
The processed mouse and human metabolome data used in this study are available as a Source Data (file name, metabolome_cems.xlsx). The mouse and human proteome data generated in this study have been deposited in jPOST repository database (project ID, JPST001807, accession ID, PXD035832; project ID, JPST002183, accession ID, PXD042807). The mouse cachectic liver DNA microarray data used in this study have been deposited in the ArrayExpress database under accession code E-MTAB-11771. Human Protein Atlas (liver expressed genes, $N = 981$ genes) reference dataset [https://www.proteinatlas.org/humanproteome/tissue/liver] downloaded in 2022 and used in this study is provided as a Source Data file (file name, reference_hpa.xlsx). The quantitative proteomic map of 28 mouse tissues using the SILAC mice re-analyzed in this study (Fig. S3d) is publicly available in PubMed Central under accession code PMC3675825 (file name, supp_M112.024919_mcp.M112.024919-2; GUID: BE9B5A17-41BF-442D-911B-DC3F6144F83B) [https://www.ncbi.nlm.nih.gov/pmc/articles/PMC3675825/]. The quantitative proteomics data of murine liver used in this study is publicly downloadable from the website of ScienceDirect through the PubMed under accession code PMID: 25470552 (table name, Table S1. Identified and Quantified Proteins in HCTs; file name, 1-s2.0 S1550413114004999-mmc2.xlsx) [https://www.sciencedirect.com/science/article/pii/S1550413114004999?via%3Dihub]. The Standards Tables of Food Composition in Japan 2015 (Seventh Revised Edition) Documentation and Table analyzed in this study (Fig. 4d–f) are openly accessible on the website of the Ministry of Education, Culture, Sports, Science and Technology, Japan [https://www.mext.go.jp/en/policy/science_technology/policy/title01/detail01/1374030.htm]. The datasets of protein abundance used in this study are publicly available in the PAXdb (mouse liver, M.musculus−Liver (Integrated)) [https://pax-db.org/dataset/10090/994751110/]; $S.$ $cerevisiae$, S.cerevisiae−Whole organism (Integrated)

[https://pax-db.org/dataset/4932/3221176528/]; *B. taurus*, B.taurus−Whole organism (Integrated) [https://pax-db.org/dataset/9913/1047829202/]; *A. thaliana*, A.thaliana−Whole organism (Integrated) [https://pax-db.org/dataset/3702/2269330278/]; human plasma excluded, H.sapiens−Plasma (Integrated) [https://pax-db.org/dataset/9606/1369124905/]. The dataset of Rossmann-like domains (RLDs) re-analyzed in this study (Fig. 4g–i) is publicly accessible in PubMed Central under accession PMC6957218 (S4 Table; GUID: 60A9D1A3-812B-4808-9F1B-834009B29B38) [https://www.ncbi.nlm.nih.gov/pmc/articles/PMC6957218/]. The datasets of crystal structure of human LDHA and NNMT presented in this study (Fig. 4g) are publicly available in the Protein Data Bank in Europe (PDBe) under the accession codes 5w8k [https://www.ebi.ac.uk/pdbe/entry/pdb/5w8k] and 2iip [https://www.ebi.ac.uk/pdbe/entry/pdb/2iip]. The dataset of secretory proteins presumed to be of liver origin used in this study (Fig. 8c, d) is publicly accessible in PubMed Central under accession PMC6723870 (file name, nutrients-11-01795-s001.zip; GUID, 5F20CF21-D0D2-4B1B-A930-DA335032CE21; table name, Table S2) [https://www.ncbi.nlm.nih.gov/pmc/articles/PMC6723870/]. Reference numbers of protein concentrations and tissue-specific gravity are freely available in the BioNumbers database under accession codes 113242 [https://bionumbers.hms.harvard.edu/bionumber.aspx?id=113242&ver=4&trm=113242&org=] and 115456 [https://bionumbers.hms.harvard.edu/bionumber.aspx?id=115456&ver=2&trm=115456&org=]. Source data are provided with this paper. The remaining data are available within the Article, Supplementary Information, or Source Data file. Source data are provided with this paper.

## Code availability

No new algorithms were developed for this manuscript. We state key R packages and their relevant function information used in this study in the Methods section. All other information related to the software used in this study is available from the corresponding authors upon reasonable request.

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

## Acknowledgements

We thank Kaori Igarashi, Kaori Saito, and Keiko Kato (Institute for Advanced Biosciences, Keio University) for CE-MS technical assistance, Kentaro Taki (Division for Medical Research Engineering, Nagoya University Graduate School of Medicine) for LC-MS technical assistance, and Kyoko Kobori and Yoshiko Goto (Aichi Cancer Center Research Institute) for technical assistance. Illustrations came from ©2016 DBCLS Togo TV and Microsoft PowerPoint. This work was supported in part by JSPS KAKEN (21K07140; to Y.K.), Suzuken Memorial Foundation (to Y.K.), Nagono Medical Foundation (to Y.K.), Daiwa Securities Health Foundation (to T.F.), JSPS KAKEN (JP26860430, JP17K15841, JP20K10463; to I.O.), Foundation for Promotion of Cancer Research (to I.O.), AMED (JP21zf0127001; to T.S.), CREST (JPMJCR2123; to T.S.), The Hori Sciences and Arts Foundation (to M.A.), Project Mirai Cancer Research Grants (to M.A.), and the Takeda Science Foundation (to M.A.).

## Author contributions

Conceptualization, Y.K.; Project administration, Y.K. and M.A.; Investigation, Y.K., E.M.S., T.F., K.S., R.K.S., and M.A.; Resources, I.O., K.N., Y.N., T.O., K.Matuso, K.Muro, M.M.T.; Data curation, Y.K. and E.M.S.; Formal analysis, Y.K.; Visualization, Y.K.; Writing—original draft, Y.K.; Writing—review and editing, Y.K. and M.A.; Supervision, T.S. and M.A.; Funding acquisition, Y.K., T.F., I.O., T.S., and M.A.

## Competing interests

The authors declare no competing interests.
