## [Peer Review file · Nature Communications]

REVIEWER COMMENTS

Reviewer #1 (Remarks to the Author):

Kojima and colleagues present a primarily proteomics-based investigation into changes in B-vitamin related enzymes in the liver of tumor-bearing animals. The authors use four different tumor models to model progressive cachexia and support their findings with analyses of blood from gastric cancer patients. The manuscript includes attractive data visualization. While the ultimate findings are likely supported, I have significant concerns that should be addressed prior to publication.

Major concerns

There is insufficient characterization on the four tumor models to demonstrate that these are bona fide cachexia models. These models are not considered well-established models in the field, and the three references provided do not provide any phenotyping of these mouse models as cachexia models. Body weight, liver weight, and the weight of a single muscle, all reported as fold control, do not meet the standard for characterization of a cachexia model. This manuscript relies heavily on assumption that the authors are modeling a progression of cachexia through the use of 4 models, but without additional characterization, it is impossible to evaluate the findings in this context.

Figure 1 does not indicate lost weight in tumor-bearing mice – it indicates that controls weighed more than the tumor-bearing mice, which is often related to failure to grow or thrive, particularly in long-term models and in models that are performed in young, rapidly growing mice. Language like reduction and lost body weight give the mistaken impression that the data presented indicate clear loss of muscle mass and body weight from mice, and that is not the case. Strain differences and differences in the length of tumor-bearing, which is not clearly reported, compound my concern about the lack of data verifying these models as cachexia models.

Much statistical power is derived from the assumption of progressive cachexia in the 4 models, and I do not find it to be effectively supported by data. It is very unclear why LNCA was not considered a cachexia model, as the mice had decreased muscle weights and body weights. This is particularly confusing as this model has recently been used as a cachexia model, with significantly more phenotyping data (PubMed ID: 34172439).

This statistical power becomes problematic for claims like in Figure 2b, which finds that muscle B6 progressively declines in cachexia – but visually, that doesn't actually appear to be the case. Similarly for muscle Gly in Figure 2e.

Without characterization, it is impossible to know that cachexia is the cause of death in SEKI mice, and therefore it is not possible to link survival to cachexia with the data presented to conclude that cachexia was prevented in supplemental figure 1. Gly could easily have simply altered the rate of tumor growth, which is not assessed in the manuscript.

The western blots in Figure 3b, claiming to support the visual changes seen in Figure 3a, are described too simply. It is fair to say that changes were confirmed – but a decrease in SDHA can't be directly tied to mitochondrial swelling. Similar issues exist with the characterization of other organelle changes.

I'm not sure that the associations between fold change and abundance of either the metabolome or enzyme proteome represented in Figure 4a are not simply reflective of the inherent probability of protein detection and likely ease of finding larger changes. Additional context for why the authors find this to be significant would have been valuable for this reviewer.

Minor concerns

The right-facing arrow in Figure 2g is confusing – I believe the objective is to report that SAM increased.

It's curious that not a single early-stage gastric cancer patient presented with cachexia, as this is unusual in my experience (supplementary table 3).

The discussion is somewhat brief and readers could use additional support in putting the findings presented here into context. In particular, if the idea is that 1C could be used as a biomarker for cachexia, that would be a helpful discussion point.

Marked changes may be an overstatement of the changes in acylation muscle in (supplementary figure 6c, line 414)

AUROC analyses would be a nice addition to Figure 8 to demonstrate the ability of C1-related metabolites to distinguish between groups of patients.

Reviewer #2 (Remarks to the Author):

In the manuscript "Decreased liver B vitamin-related enzymes as a metabolic hallmark of cancer cachexia", Kojima et al. describe the metabolic landscape of liver and muscle of five mouse models related to cancer or metabolic dysfunction. Using metabolome profiling, proteomics, transcriptomics, and post-translational modification profiling, they found reduced metabolites of the B vitamin class and one-carbon metabolism, enzymes linked to these metabolites, and protein malonylation in livers of mice with cachexia. They also recapitulate some of these findings in plasma of patients with gastric cancer.

Overall, this is an excellently executed and written study presenting novel data on liver metabolism in cancer cachexia. The methods section is written in a way that allows recapitulation. The authors use an interesting approach to integrate multiple omics datasets, and the inclusion of publicly available datasets supports the own data and data interpretation. Conclusions are mostly supported by the data, and the authors clearly state when they speculate on findings.

While I am overall supportive of this study, including data or comments on some aspects would further improve the story:

1) Mouse models used in the study

Including a starvation control, the authors could demonstrate that the effects caused by cancer-related weight loss are not related to simply anorexia of the tumor-bearing mice. This is an elegant control, although the rationale for 48h starvation should be given, as this is an extremely long starvation phase for a mouse, and does not represent the anorexia that typically occurs in cachexia. For better comparison and to assess the degree of anorexia in the used mouse models, food intake over time should be recorded for the cancer models.

No justification on the use of mouse models is given, although these are neither commonly used mouse models for cancer cachexia, nor do they represent a specific tumor type. Melanoma is typically associated with a lower risk of cachexia. A better justification for the use of mouse models should be given in the main text.

Cancer cachexia is defined not only by loss of body weight and muscle mass, but also by adipose tissue loss and systemic inflammation. Since these are not commonly used mouse models, it will be important to include a more detailed phenotyping of the cachexia pathology including weight loss, fat mass, time curves (time to disease), inflammatory markers.

2) Linking presented data with known cachexia profiles

How does the presented data correlate with molecular markers of cancer cachexia? Can molecular markers of muscle atrophy (for instance MuRF1, atrogin, activation of autophagy or UPS) be found in the current data, and how are they linked with the presented data on B vitamins and one carbon metabolism? This data should be included in the manuscript.

3) Pharmacological intervention

FK866 and 5FU (sup fig 2) treatment was only used on cancer-free wildtype mice, so the relevance of these treatments is somehow limited and unclear. Treatment of cancer bearing mice is therefore missing. 5FU is also used as chemotherapeutic reagent and has multiple effects not related to the pathways presented here, so its relevance for the current study does not become so clear, especially since it also induced cachexia in mice (e.g. PMID: 33364975). This needs to be addressed in order to contextualize this data better.

4) Liver ultrastructure

The data on liver ultrastructure presented in Fig 3a,b is a bit too thin to allow definite conclusions. Data from Fig. 3a (i.e. the features highlighted in EM pictures) should be quantified from all mice instead of just showing one representative figure. Are the ultrastructural changes also occurring in mouse models of cancer without cachexia?

5) Patient data

It is unclear why gastric cancer patients were used in the current study, as it does not match with the mouse models. To which degree do mechanical problems (problems in swallowing, vomiting) contribute to the cachexia in these patients? The patient data is partly inconsistent with the mouse data, and the sample size is small. The findings are still relevant, but these should be indicated as limitations of the study. How do the presented data relate to other published metabolomics datasets from patients with cachexia? (e.g. PMID: 29152916)

Additional minor points:

- Were samples taken in fasted state or was otherwise controlled for food intake? This is relevant for interpretation of results
- Only female mice were used for the study. This needs to be commented on and a rationale should be given.
- Line 74: "We employed five mouse models for this study to identify metabolic features of cancer cachexia": It is incorrect to state that 5 cancer cachexia models were used. One of them was starvation. MEWO did not lose weight so is also no cachexia model.
- Typo in figure legend 1e (blue)
- Fig. 3c, typo in legend of x axis ("control mice")
- Fig. 3 legend: "Mouse folate, biotin, and B12 values were collected from previous reports." These previous reports should be cited here
- Fig. 5b, c: axis labels too small
- Typo in Fig 5e (in figure, proteins) and legend of 5d (numbers next to arrows)
- Lines 171 to 181 all these correlations and "links" about pathways are really unclear and should be explained better
- Sup Fig 2b seems to contain duplicated values
- Line 392, what does it mean, "decline in an abundance-dependent manner"? Please re-write or explain better
- Line 489 the word "liver" is missing

Reviewer #3 (Remarks to the Author):

The manuscript "Decreased liver B vitamin-related enzymes as a metabolic hallmark of cancer cachexia" by Kojima et al. identified a mechanistic basis for two mouse models of cachexia through an integrated metabolomics/proteomics analyses. They suggest that although cachexia results in muscular defects, that cachexia begins with metabolite/enzyme changes in the liver, not the skeletal muscle. This point in itself is not novel, but the data and mechanism proposed is interesting and likely to be correct, at least for these two mouse models of cachexia. The experimental design is thorough

on the technical level – particularly proteomic and metabolomic based mass spec and relevant analyses. The observation that cofactor metabolites are stoichiometrically 1:1 related to cofactor enzymes is surprising and to my knowledge, novel, if true. The observations about malonylation of liver proteins are also striking and translational to human samples.

Major points:

1. Do the authors have any insight into which proteins are differentially malonylated between the mouse and human studies? (Similar to Figure 8a but for proteins not metabolites).

NCOMMS-22-28618A-Z: Response to Reviewer Comments

Reviewer #1 (Remarks to the Author):

Kojima and colleagues present a primarily proteomics-based investigation into changes in B-vitamin related enzymes in the liver of tumor-bearing animals. The authors use four different tumor models to model progressive cachexia and support their findings with analyses of blood from gastric cancer patients. The manuscript includes attractive data visualization. While the ultimate findings are likely supported, I have significant concerns that should be addressed prior to publication.

We appreciate the reviewer's thoughtful suggestions and comments, which have strengthened the manuscript.

Major concerns

There is insufficient characterization on the four tumor models to demonstrate that these are bona fide cachexia models. These models are not considered well-established models in the field, and the three references provided do not provide any phenotyping of these mouse models as cachexia models. Body weight, liver weight, and the weight of a single muscle, all reported as fold control, do not meet the standard for characterization of a cachexia model. This manuscript relies heavily on assumption that the authors are modeling a progression of cachexia through the use of 4 models, but without additional characterization, it is impossible to evaluate the findings in this context.

This was a plausible concern. In response, we have provided extensive phenotypic data demonstrating that our models develop bona fide cancer cachexia. Specifically, we created a new supplementary figure (Fig. S1) containing Kaplan-Meier survival data, body weight time-series data, and food intake data, qPCR data of muscle atrophy markers, and ELISA (enzyme-linked immunosorbent assay) data of cachexia-related factors in plasma. We have also summarized these phenotypic data in Fig 1a and edited the main text accordingly (Page 5, line 81). "Analyses of survival, body weight, food intake, body composition, muscle atrophy markers, and plasma proteins indicated that SEKI, LNCA, and CRCA undergo body weight loss with muscle wasting and reduced food intake, phenotypes characteristic of cancer cachexia (Fig. S1a-e)."

Figure 1 does not indicate lost weight in tumor-bearing mice – it indicates that controls weighed more than the tumor-bearing mice, which is often related to failure to grow or thrive, particularly in long-term models and in models that are performed in young, rapidly growing mice. Language like reduction and lost body weight give the mistaken impression that the data presented indicate clear loss of muscle mass and body weight from mice, and that is not the case. Strain differences and differences in the length of tumor-bearing, which is not clearly reported, compound my concern about the lack of data verifying these models as cachexia models.

We thank the reviewer for his/her insightful comments. We think that the new time-series data on body weight change will alleviate the reviewer's concerns (Fig. S1b). Body

weights of SEKI melanoma xenograft model (SEKI) and genetically engineered mouse models of lung cancer (LNCA) and colorectal cancer (CRCA) peaked and then began to decrease, instead of slowly increasing or stabilizing. Therefore, we think that the expressions “body weight reduction” and “loss of body weight” are acceptable. Although we were unable to obtain time-series data on muscle weight, we think that increased levels of muscle atrophy markers (Fig. S1d) support the use of “muscle reduction” or “loss of muscle.”

Much statistical power is derived from the assumption of progressive cachexia in the 4 models, and I do not find it to be effectively supported by data. It is very unclear why LNCA was not considered a cachexia model, as the mice had decreased muscle weights and body weights. This is particularly confusing as this model has recently been used as a cachexia model, with significantly more phenotyping data (PubMed ID: 34172439).

Based on the extensive phenotypic data above, we now agree with the reviewer’s comment that the LNCA model develops cachexia (Fig. S1a-e). Therefore, we have changed our dichotomous classification of models to “cachectic versus non-cachectic models” in the revised manuscript. On the other hand, phenotypic data indicate different severities of cancer cachexia symptoms among our models. Specifically, evaluation of these mouse models with changes in five parameters, body weight, muscle weight, muscle atrophy marker levels, plasma serum amyloid A (SAA) level, and plasma albumin level, shows a trend from MEWO, LNCA, and SEKI to CRCA (Fig. 1a, b, Fig. S1b, c, d, e). Metabolome volcano plots support the escalation trend from MEWO to CRCA (Fig. 1c), and principal component analyses of metabolomic data, and phenotypic data suggest a potential boundary between MEWO/LNCA and SEKI/CRCA (Fig. 1d; dashed lines). Hepatic ultrastructural data also support the boundary (Fig. 3a-c). Based on these additional findings, we have postulated a cachexia severity gradient from MEWO to CRCA and a phenotypic boundary between LNCA and SEKI in the revised manuscript (Fig 1b). Accordingly, we have modified the description regarding cachexia/non-cachexia dichotomy throughout the manuscript (e.g., Page 5, line 73).

This statistical power becomes problematic for claims like in Figure 2b, which finds that muscle B6 progressively declines in cachexia – but visually, that doesn’t actually appear to be the case. Similarly for muscle Gly in Figure 2e.

We agree that the statistical results of muscle data do not match the graphical appearance. We suspect that MEWO or CRCA acts as an outlier for our statistical method that is vulnerable to outliers (the Jonckheere trend one-sided test). Therefore, we have modified the text as follows to clarify that muscle B6 and Gly levels do not reflect cachexia severity.

Page 8, line 126; “hepatic NIA levels”

Page 8, line 128; “vitamin B6 (B6) levels in liver”

Page 9, line 141; “muscle and hepatic betaine and hepatic Gly from MEWO to CRCA”

In addition, we have changed the P value fonts to grey when the P value was greater than 0.0001 (Fig. 2a, b, e, f).

Without characterization, it is impossible to know that cachexia is the cause of death in SEKI mice, and therefore it is not possible to link survival to cachexia with the data presented to conclude that cachexia was prevented in supplemental figure 1. Gly could easily have simply altered the rate of tumor growth, which is not assessed in the manuscript.

As requested, we have provided data showing the effects of glycine administration on body weight, food intake, and tumor growth (Fig. S3c). Although the data suggest that the effect of Gly on survival is not likely due to reduced tumor growth, further study is required to determine the underlying mechanism. Accordingly, we have added the following sentence in the main text.

Page 9, line 144; “However, it caused no significant changes in body weight, food intake, or tumor growth, and the underlying mechanism remains unclear (Fig. S3c).”

”

The western blots in Figure 3b, claiming to support the visual changes seen in Figure 3a, are described too simply. It is fair to say that changes were confirmed – but a decrease in SDHA can't be directly tied to mitochondrial swelling. Similar issues exist with the characterization of other organelle changes.

We have removed western blotting data (Fig. 3b of the original manuscript) because we think that our proteomic data visualization suffices to demonstrate hepatic changes associated with cancer cachexia at the protein level (Fig. 3d). Instead, in response to Reviewer 2's comment, we have provided additional electron micrographs to better depict organelle changes in the liver (Fig. 3a-c).

I'm not sure that the associations between fold change and abundance of either the metabolome or enzyme proteome represented in Figure 4a are not simply reflective of the inherent probability of protein detection and likely ease of finding larger changes. Additional context for why the authors find this to be significant would have been valuable for this reviewer.

Fig. 4a (original manuscript)

Fig. 4a (original manuscript) visually supports two hypothetical associations: between abundance and fold-change value and between proteome and metabolome. Because enzymes and metabolites share an extremely unequal abundance distribution pattern in tissues (Milo R. and Phillips R., *Cell Biology by the Numbers* (Garland Science, 2016)), we inferred an association between abundance and fold-changes. We also suspected an association between enzyme and metabolite abundances and changes, because enzymes and metabolites may be highly bound in tissues (PMID: 6054167). Therefore, we considered Fig. 4a a good infographic in the original manuscript. However, we understand the reviewer's concern. We are unable to accurately assess and eliminate the contribution of abundance-related noise in the original Fig. 4a. Also, our original manuscript did not clearly describe the two hypothetical associations mentioned above. Accordingly, we have replaced Fig. 4a with volcano plots and revised the text to introduce our strategy more clearly (Fig. 4a; Page 14, line 231).

Minor concerns

The right-facing arrow in Figure 2g is confusing – I believe the objective is to report that SAM increased.

In accordance with this comment, we have corrected the hepatic SAM arrow from right-facing to up-facing (Fig. 2g).

It's curious that not a single early-stage gastric cancer patient presented with cachexia, as this is unusual in my experience (supplementary table 3).

We apologize for the inadequate explanation of "NA" in Supplementary Table 3. This NA did not mean that the patients did not have cachexia, but that their quick access medical records for diagnosing cachexia were NOT AVAILABLE. We have now performed a more comprehensive review of medical records and updated data regarding the presence or absence of cachexia in patients with early gastric cancer (Table S3). Accordingly, when calculating fold changes in metabolite data for gastric cancer patients, we excluded two early gastric cancer patients diagnosed with cachexia from the control group. As a result, Figs. 8a and b have been slightly changed. We have also modified the text (Page 25, line 433; 15 metabolites). However, these corrections do not affect our conclusions.

The discussion is somewhat brief and readers could use additional support in putting the findings presented here into context. In particular, if the idea is that 1C could be used as a biomarker for cachexia, that would be a helpful discussion point.

We have added a paragraph on biomarkers of cancer cachexia in the Discussion (Page 28, line 498).

Marked changes may be an overstatement of the changes in acylation muscle in (supplementary figure 6c, line 414)

We agree with the reviewer's comment and have changed "marked alterations" to "alterations" (Fig S8c; page 23, line 400).

AUROC analyses would be a nice addition to Figure 8 to demonstrate the ability of C1-related metabolites to distinguish between groups of patients.

We have added a figure panel of ROC curve analyses (Fig. S9a) and the following sentence (Page 25, line 438). "ROC curve analysis suggested that a combination of choline and Trp predicts cachexia more successfully than CRP and albumin (Fig. S9a)."

Reviewer #2 (Remarks to the Author):

In the manuscript "Decreased liver B vitamin-related enzymes as a metabolic hallmark of cancer cachexia", Kojima et al. describe the metabolic landscape of liver and muscle of five mouse models related to cancer or metabolic dysfunction. Using metabolome profiling, proteomics, transcriptomics, and post-translational modification profiling, they found reduced metabolites of the B vitamin class and one-carbon metabolism, enzymes linked to these metabolites, and protein malonylation in livers of mice with cachexia. They also recapitulate some of these findings in plasma of patients with gastric cancer.

Overall, this is an excellently executed and written study presenting novel data on liver metabolism in cancer cachexia. The methods section is written in a way that allows recapitulation. The authors use an interesting approach to integrate multiple omics datasets, and the inclusion of publicly available datasets supports the own data and data interpretation. Conclusions are mostly supported by the data, and the authors clearly state when they speculate on findings.

While I am overall supportive of this study, including data or comments on some aspects would further improve the story:

We sincerely appreciate the reviewer's positive comments and invaluable suggestions that have improved our manuscript.

1) Mouse models used in the study

Including a starvation control, the authors could demonstrate that the effects caused by cancer-related weight loss are not related to simply anorexia of the tumor-bearing mice. This is an elegant control, although the rationale for 48h starvation should be given, as this is an extremely long starvation phase for a mouse, and does not represent the anorexia that typically occurs in cachexia. For better comparison and to assess the

degree of anorexia in the used mouse models, food intake over time should be recorded for the cancer models.

We thank the reviewer for drawing attention to the importance of anorexia in cancer cachexia. We have added daily food intake data in the revised manuscript (Fig. S1b).

Figure 1 for peer review

Starvation for 48 hours was not intended to mimic anorexia associated with cachexia. We prioritized body compositional changes over food intake when establishing a starvation protocol to help identify characteristic changes in cancer cachexia. As some 24-hour fasted mice (STRV 24hrs) maintained substantial muscle and white adipose tissue weights (Figure 1 for peer review above), we preferred to use the 48-hour starvation model (STRV 48hrs). For readers, we have described the rationale for adopting this 48-hour starvation protocol in the Methods (Page 31, line 550). In addition, we have added the following phrase to the main text.

Page 5, line 80; “produce body compositional and metabolic changes...”

No justification on the use of mouse models is given, although these are neither commonly used mouse models for cancer cachexia, nor do they represent a specific tumor type. Melanoma is typically associated with a lower risk of cachexia. A better justification for the use of mouse models should be given in the main text.

Cancer cachexia is defined not only by loss of body weight and muscle mass, but also by adipose tissue loss and systemic inflammation. Since these are not commonly used mouse models, it will be important to include a more detailed phenotyping of the cachexia pathology including weight loss, fat mass, time curves (time to disease), inflammatory markers.

In accordance with the reviewer's incisive comments, we have added extensive

phenotypic data, including quantification of adipose tissues and inflammatory markers (Fig. S1, Kaplan-Meier survival data, body weight, food intake data, body composition, muscle qPCR data, and plasma ELISA data). We believe that these additional data conclusively demonstrate that our mouse models are bona fide cancer cachexia models.

This research aims to better understand cachexia-related metabolic features conserved across different cancer types, instead of investigating cancer-type-specific characteristics. We selected SEKI, LNCA, and CRCA, because these models reproducibly develop cachectic phenotypes in our hands. For better justification, we have added the following phrase to the text (Page 5, line 75); “while avoiding results limited to a particular model or cancer type...”

We agree with the reviewer that the risk of cancer cachexia in malignant melanoma patients is relatively low. However, ~20,000 melanoma patients are estimated to develop cancer cachexia annually in the United States (PMID: 30920776), which is close to the number of US gastric cancer patients with cachexia symptoms. Furthermore, melanoma cachexia state is reported to be a significant inhibitor factor in immune-checkpoint inhibitor therapy against melanoma (PMID: 29891725). We hope that our findings will benefit future melanoma patients.

2) Linking presented data with known cachexia profiles

How does the presented data correlate with molecular markers of cancer cachexia? Can molecular markers of muscle atrophy (for instance MuRF1, atrogin, activation of autophagy or UPS) be found in the current data, and how are they linked with the presented data on B vitamins and one carbon metabolism? This data should be included in the manuscript.

We thank the reviewer for the constructive suggestion. We have added quantitative PCR data for three muscle atrophy-related genes, *MuRF1(Trim63)*, *Atrogin-1(Fbxo32)*, and *Fkhr (Foxo1)* (Fig. S1d). Our data confirmed muscle atrophy in LNCA, SEKI, CRCA, and STRV. We also noticed that mRNA levels of *MuRF1* and *Atrogin-1* in the five models, including STRV, appeared to be inversely correlated with severity of muscle loss and muscle SAM levels. We have added the following phrase to the main text.

Page 9, line 154; “and expression of muscle atrophy markers (Fig. S1c, d)”

3) Pharmacological intervention

FK866 and 5FU (sup fig 2) treatment was only used on cancer-free wildtype mice, so the relevance of these treatments is somehow limited and unclear. Treatment of cancer bearing mice is therefore missing. 5FU is also used as chemotherapeutic reagent and has multiple effects not related to the pathways presented here, so its relevance for the current study does not become so clear, especially since it also induced cachexia in mice (e.g. PMID: 33364975). This needs to be addressed in order to contextualize this data better.

We should have explained the specific purpose of pharmacological experiments using FK866 and 5FU more clearly. 5FU is indeed a typical chemotherapy drug and FK866 was once considered a candidate anticancer agent. However, in this study, we performed these pharmacological experiments simply to disturb the NAD synthesis pathway or C1 metabolic pathway, and we did not intend to examine their anticancer effects. We have revised the text to clarify the limited purpose of the FK866 and 5FU experiments.

Page 10, line 163; “We then explored a metabolic link between the two pathways using anti-tumor compounds as simple metabolic disturbing chemicals”

4) Liver ultrastructure

The data on liver ultrastructure presented in Fig 3a,b is a bit too thin to allow definite conclusions. Data from Fig. 3a (i.e. the features highlighted in EM pictures) should be quantified from all mice instead of just showing one representative figure. Are the ultrastructural changes also occurring in mouse models of cancer without cachexia?

To address the reviewer’s concern, we have provided a series of ultrastructural images of the liver for all models with quantification data of highlighted features (Fig. 3a-c). On the other hand, we have removed the western blotting data as we think that our proteomic data suffices to demonstrate hepatic changes associated with cancer cachexia at the protein level. Our new data suggest that ultrastructural changes in the liver are associated with the severity of cancer cachexia-related phenotypes.

5) Patient data

It is unclear why gastric cancer patients were used in the current study, as it does not match with the mouse models. To which degree do mechanical problems (problems in swallowing, vomiting) contribute to the cachexia in these patients? The patient data is partly inconsistent with the mouse data, and the sample size is small. The findings are still relevant, but these should be indicated as limitations of the study. How do the presented data relate to other published metabolomics datasets from patients with cachexia? (e.g. PMID: 29152916)

We selected and examined gastric cancer patients for this project for a practical reason: gastric cancer is common among East Asians, and our colleagues at Aichi Cancer Center Hospital have good access to severe cachexia patients with gastric cancer. Since this project aims to extract common metabolic phenotypes conserved across different cancer types (and species), rather than to investigate cancer-type-specific characteristics, we do not see a problem in studying gastric cancer patients, who very often develop severe cachexia.

We added a table row on gastrointestinal obstruction, one of the mechanical problems, and noted the rationale for selecting gastric cancer samples and the limitations of our clinical data in the main text.

Page 25, line 429; “To address whether the above-mentioned candidate hallmark of **severe** cancer cachexia is reflected in the blood and to assess its clinical relevance, we performed metabolomic analyses using blood samples from **CRCA mice** and from gastric cancer patients, who frequently develop **severe** cachexia (Table S3).”

We thank the reviewer for mentioning this interesting paper (PMID: 29152916, Yang QJ. et al., 2017, Journal of Cachexia Sarcopenia Muscle). We notice some discrepancies between Yang’s metabolomic data and ours. Clearly, more extensive blood metabolomic analysis of cancer cachexia needs to be conducted to provide an integrated interpretation of our findings and data from others. We have pointed out the limitation of our clinical data in the Discussion (Page 28, line 499). “**Although the small number of clinical samples and analysis of a single cancer type are limitations of this study, our findings on choline and tryptophan agree with previous reports.**”

Additional minor points:

-Were samples taken in fasted state or was otherwise controlled for food intake? This is relevant for interpretation of results

We have added the following sentence to the Methods (Page 33, line 577); “**We collected all mouse samples from ad-lib fed conditions, except the starvation group.**”

- Only female mice were used for the study. This needs to be commented on and a rationale should be given.

We thank the reviewer for raising an important issue. The revised manuscript delivers two critical findings based on sex-based analyses. First, we did not observe biologically significant sex differences in body compositional changes (Fig.S2). Second, hepatic SEKI and CRCA proteomic data also showed no substantial sex differences that challenge our main findings (Fig. S7a, b). Accordingly, we have added the following sentences to the Results.

Page 6, line 93; “**We observed no substantial sex differences in survival time or body compositional changes (Fig. S2a, b).**”

Page 20, line 347; “**There seemed to be no substantial sex-related differences in collective behaviors of these enzymes in SEKI and CRCA livers (Fig. S7a, b).**”

- Line 74: “We employed five mouse models for this study to identify metabolic features of cancer cachexia”: It is incorrect to state that 5 cancer cachexia models were used. One of them was starvation. MEWO did not lose weight so is also no cachexia model.

We have corrected the sentence as follows:

Page 5, line 74; “We employed **multiple** mouse models for this study to identify metabolic features of cancer cachexia...”

- Typo in figure legend 1e (blue)

We thank the reviewer for pointing out the spelling error. We have corrected it.

- Fig. 3c, typo in legend of x axis (“control mice”)

We have fixed the typo (Fig. 3d).

- Fig. 3 legend: “Mouse folate, biotin, and B12 values were collected from previous reports.” These previous reports should be cited here

We have added the reference information to the figure legend (Fig. 4d).

- Fig. 5b, c: axis labels too small

We have increased the font size of axis labels.

- Typo in Fig 5e (in figure, proteins) and legend of 5d (numbers next to arrows)

Thanks again for pointing out the spelling errors. We have corrected them.

- Lines 171 to 181 all these correlations and “links” about pathways are really unclear and should be explained better

We have simplified the description of the correlation among metabolites as follows (Page 10, line 175). “In addition, our integrated pooled mouse liver metabolomic data showed that SAM and its derivatives, 1-methyl nicotinamide (MNA) and putrescine, were negatively related to NIA and B6 (Fig. S3f, g).”

- Sup Fig 2b seems to contain duplicated values

We have removed the duplicated values (Fig. S3f).

- Line 392, what does it mean, “decline in an abundance-dependent manner”? Please re-write or explain better

We have rewritten the sentence as follows (Page 22, line 378). “SEKI and CRCA livers showed concentration-weighted downregulation of Gly-related enzymes, but not SAM-related enzymes (Fig. 6c).”

- Line 489 the word “liver” is missing

We thank the reviewer for pointing out the missing word. We have corrected the sentence as follows. “These findings lead us to speculate that the cachectic liver may not be...” (Page 27, line 478).

Reviewer #3 (Remarks to the Author):

The manuscript “Decreased liver B vitamin-related enzymes as a metabolic hallmark of cancer cachexia” by Kojima et al. identified a mechanistic basis for two mouse models

of cachexia through an integrated metabolomics/proteomics analyses. They suggest that although cachexia results in muscular defects, that cachexia begins with metabolite/enzyme changes in the liver, not the skeletal muscle. This point in itself is not novel, but the data and mechanism proposed is interesting and likely to be correct, at least for these two mouse models of cachexia. The experimental design is thorough on the technical level – particularly proteomic and metabolomic based mass spec and relevant analyses. The observation that cofactor metabolites are stoichiometrically 1:1 related to cofactor enzymes is surprising and to my knowledge, novel, if true. The observations about malonylation of liver proteins are also striking and translational to human samples.

We appreciate the reviewer's positive evaluation of our findings.

Major points:

1. Do the authors have any insight into which proteins are differentially malonylated between the mouse and human studies? (Similar to Figure 8a but for proteins not metabolites).

We thank the reviewer for the critical question. We have created a new supplementary figure (Fig. S10a, b) and added the description in the Results as follows (Page 26, line 456).

“Furthermore, analysis of blood samples from advanced gastric cancer cachexia patients revealed that most malonylated and acetylated proteins commonly detected in mouse and human plasma are primarily liver-derived (Fig. S10a, b) and that some cases show a tendency toward decreased malonylation and acetylation, similar to SEKI and CRCA (Fig. 8e).”

REVIEWER COMMENTS

Reviewer #1 (Remarks to the Author):

Thanks to the authors for their thoughtful consideration of my concerns with their previous manuscript. The current version has addressed many of my concerns, including phenotyping the cachexia models to ensure that they actually caused cachexia. However, I have some additional concerns that either remain from the previous review or have arisen from changes in the manuscript.

The authors should add recent work on the role of NAD in cancer cachexia to their reference list (PMID: 37012289). The publication of this paper does not in any way decrease the significance of this work, and likely increases the importance of the present work.

In Figure 1D, it is unclear to this reviewer if the starvation (STRV) was taken into account with linear SVM, because while STRV appears with the severe cachexia in the muscle blot, STRV clusters with MEWO and LNCA, the no/low cachexia models, in liver and body composition. If this is correct, then the text in line 102 about a marked boundary is not accurate, as the STRV is clustering with different groups. Text should be edited to reflect the data more accurately. Additionally, the data used to derive the "body compositional" PCA plot should be more clearly described in the legend, as it is not obvious to this reviewer. Is the data reported in supplemental figure 1 of tissue masses?

I remain concerned about the statistical methods used to achieve significance in Figure 2a, in using a trend test based upon an artificial continuum of cachexia. The data in this figure appear again in supplemental Figure 3e, where they are analyzed by a t-test and found to be not statistically significant. No, being statistically significant doesn't mean everything, but transparency in the data matter, and I believe that additional transparency is warranted in the text that more conventional methods of assessing the data do not find statistical significance. I find this to be particularly relevant for things like liver NAM and liver B6, where a significant portion of the power necessary to find significance in the trend is derived from increases in the MEWO model.

Based upon the data provided, the assumption that SEKI mice died because of their cachexia is not supported and therefore the text surrounding Figure S3a is not factually correct. The conclusion cannot be drawn that B vitamins alone are insufficient to improve cachexia with data shown, only that they do not improve overall survival.

The text surrounding the comparison between early and late gastric cancer patients (line 434) could be more clear – as currently written, it is unclear what groups were compared.

Similarly, the text surrounding malonylation/acetylation after 6 months in patients should be clearer – at current, it is difficult to appreciate that these are serial blood samples from the same patient, with comparisons made based upon weight loss over that 6-month period.

Reviewer #2 (Remarks to the Author):

The authors have addressed my concerns and have significantly improved their manuscript, in particular in regard to better describing the mouse models that were used in the study. This was a very crucial point.

Since the first revision of the manuscript, a new study has appeared describing NAD⁺ metabolism and niacin in the cachexia context (PMID: 37012289) which should be critically addressed here, since the findings therein are important for the current study.

Please note that there is a typo in headline of figure 1d (live instead of liver)

Reviewer #3 (Remarks to the Author):

The author's addressed my main point and the rebuttal to the other Reviewer's has strengthened the manuscript.

REVIEWER COMMENTS

Reviewer #1 (Remarks to the Author):

Thanks to the authors for their thoughtful consideration of my concerns with their previous manuscript. The current version has addressed many of my concerns, including phenotyping the cachexia models to ensure that they actually caused cachexia. However, I have some additional concerns that either remain from the previous review or have arisen from changes in the manuscript.

→ We thank the reviewer again for insightful comments and constructive suggestions on our manuscript.

The authors should add recent work on the role of NAD in cancer cachexia to their reference list (PMID: 37012289). The publication of this paper does not in any way decrease the significance of this work, and likely increases the importance of the present work.

We are grateful that the reviewer brought this significant work to our attention. We have cited the paper and also several additional papers on hepatic NAD metabolism and cancer cachexia.

Page 7, line 118; PMID: 27153497, PMID: 32298240, PMID: 32599075, PMID: 34940589, and PMID: 37012289.

Page 11, line 191; PMID: 9703363, and PMID: 35705545

In Figure 1D, it is unclear to this reviewer if the starvation (STRV) was taken into account with linear SVM, because while STRV appears with the severe cachexia in the muscle blot, STRV clusters with MEWO and LNCA, the no/low cachexia models, in liver and body composition. If this is correct, then the text in line 102 about a marked boundary is not accurate, as the STRV is clustering with different groups. Text should be edited to reflect the data more accurately. Additionally, the data used to derive the “body compositional” PCA plot should be more clearly described in the legend, as it is not obvious to this reviewer. Is the data reported in supplemental figure 1 of tissue masses?

→ We thank the reviewer for pointing out our misleading description. In fact, the SVM algorithm was applied to datasets that did not include STRV data. We have modified the main text and figure legends to accurately describe the SVM procedures.

Page 6, line 103; other cancer models (MEWO and LNCA)

Page 12, line 199; from other cancer models (MEWO and LNCA)

Page 13, line 218; severe cachexia models (SEKI and CRCA) from other cancer models (MEWO and LNCA)

Page 74, line 1297; other cancer models (MEWO and LNCA)

We also apologize for the inadequate explanation of body compositional data used in Fig. 1d. We have added the following phrase to the figure legend.

Page 74, line 1293; body compositional data (changes in body weight and weights of gastrocnemius muscles, anterior tibialis muscles, soleus muscles, liver, subcutaneous and visceral white adipose tissue, brown adipose tissue, kidney,

and spleen).

I remain concerned about the statistical methods used to achieve significance in Figure 2a, in using a trend test based upon an artificial continuum of cachexia. The data in this figure appear again in supplemental Figure 3e, where they are analyzed by a t-test and found to be not statistically significant. No, being statistically significant doesn't mean everything, but transparency in the data matter, and I believe that additional transparency is warranted in the text that more conventional methods of assessing the data do not find statistical significance. I find this to be particularly relevant for things like liver NAM and liver B6, where a significant portion of the power necessary to find significance in the trend is derived from increases in the MEWO model.

→ The reviewer voiced serious concerns over our dependence on unconventional statistical methods and our imaginary continuum of cachexia severity across four different models. To address these concerns, we have made significant changes to the manuscript.

First, we replaced the Jonckheere-Terpstra trend test with the more conventional non-parametric Steel's multiple comparison test for LNCA, SEKI, and CRCA groups (Table S3). We excluded the MEWO group from the multiple comparison tests to avoid the undesirable effects of the model mentioned by the reviewer. Based on phenotypic data (Fig. 1a, S1), we consider LNCA as controls a priori. In addition, especially for liver NIA (niacin) and liver B6 (vitamin B6), we performed parametric Dunnett's multiple comparison tests for the same three groups (LNCA, SEKI, and CRCA) (Fig 2b; Table S4). The Dunnett's 95% confidence intervals supported a decreasing trend for changes of hepatic NIA and B6 from LNCA to CRCA (Table S4). For a more transparent statistical interpretation of the metabolomic data, we also provided the results of Welch's 2-tailed t-test with the respective control group for all experimental groups (Table S2). In addition, we revised the main text to help readers understand what is supported by statistical evidence. Importantly, these changes in statistical analyses did not affect our main findings and conclusions.

Page 7, line 117; NAD was decreased in muscle and liver of our four cancer-bearing models (Fig. 2a; Table S2). Notably, we found a downward trend in NAD from LNCA to CRCA (Table S3). In addition, liver metabolome data showed a decrease in NADP, an NAD derivative, and nicotinamide (NAM), the NAD precursor in severe cachexia models, SEKI and CRCA (Fig2a; Table S2). These data suggest that this alteration of NAD metabolism is present in muscle and liver of cancer-bearing mice and is associated with cachexia severity.

Page 8, line 126; When we summed NAD-related metabolites and regarded the aggregate value as NIA, we also found a significant downward trend in hepatic NIA levels from LNCA to CRCA (Fig. 2b; Table S4). A similar downward trend was found in vitamin B6 (B6) levels in liver (Fig. 2c; Table S4), where B6-related metabolites are primarily metabolized.

Page 9, line 141; Our metabolomic data showed a decrease in hepatic betaine and Gly in severe cachexia models, SEKI, and CRCA (Fig. 2e; Table S2).

Page 9, line 154; While SAM levels tended to decrease in muscle along with

increasing severity of muscle loss and expression of muscle atrophy markers (Fig. S1c, d), they tended to be maintained in livers of SEKI and CRCA mice (Fig. 2f; Table S2).

Page 11, line 176; Treatment of wild-type C57BL/6N mice with FK866, a highly selective inhibitor of nicotinamide phosphoribosyltransferase (NAMPT) that blocks NAD production³⁰, not only reduced muscle and liver NAD and NIA levels, but also affected C1 metabolism, causing a decrease in the liver Gly level and an increase in the liver SAM/SAH ratio (Fig. S4a; Table S2). Moreover, FK866 treatment significantly reduced vitamin B6, suggesting a close metabolic connection between NIA and vitamin B6 (Fig. S4a; Table S2).

Page 11, line 185; 5FU treatment also reduced muscle NIA levels of wild-type C57BL/6N mice (Fig. S4a; Table S2). Notably, these two drugs, FK866 and 5FU, increased the SAM/SAH ratio in liver, but not in muscle (Fig. S4a; Table S2). In addition, our integrated pooled mouse liver metabolomic data showed that SAM and its derivatives, 1-methyl nicotinamide (MNA) and putrescine, were negatively related to NIA and B6 (Fig. S4b, c; Table S2, S3).

Second, to address and alleviate the reviewer's concern regarding our hypothetical continuum of cachexia severity across four cancer models, we conducted additional analysis of a CRCA dataset, which was heterogeneous in regard to cachexia severity because it included CRCA mice under 10 weeks of age that exhibited less severe cachexia (Fig. S3e).

Fig. S3e

Notably, we were able to confirm linear relationships between changes in metabolite levels and muscle weight changes in the CRCA-only dataset as well, supporting our hypothesis that levels of NAD, NIA, B6, betaine, and glycine, and the SAM/SAH ratio change progressively with cachexia severity. We have documented the new findings in the main text as follows.

Page 10, line 158; Our integrated analysis of metabolomic datasets from three models (LNCA, SEKI, and CRCA), highlighted correlations between cachexia severity and specific metabolite changes (NAD, NIA, B6, betaine, glycine, and SAM). To further confirm these correlations between cachexia severity and metabolites, we analyzed a CRCA dataset, which exhibited heterogeneity of cachexia severity, because it included mice under 10 weeks of age that did not develop severe cachexia. Scatterplot analysis showed linear relationships between an indicator of cancer severity, muscle weight change, and changes in metabolites (NAD, NIA, B6, betaine, and glycine levels, and SAM/SAH ratio) in CRCA, reconfirming that these metabolic changes are correlated with cachexia severity (Fig. S3e).

Based upon the data provided, the assumption that SEKI mice died because of their cachexia is not supported and therefore the text surrounding Figure S3a is not factually correct. The conclusion cannot be drawn that B vitamins alone are insufficient to improve cachexia with data shown, only that they do not improve overall survival.

→ We agree with the reviewer's comment and revised the text surrounding Figure S3a as follows, removing the phrase "indicating that B vitamins alone are insufficient to improve SEKI cachexia":

Page 8, line 130; To address the possible effect of B vitamin supplements on the survival of severe cachexia models, we administered a high-dose B vitamin cocktail, including NIA and B6, to SEKI mice²². However, the cocktail showed no effects on their survival (Fig. S3a).

The text surrounding the comparison between early and late gastric cancer patients (line 434) could be more clear – as currently written, it is unclear what groups were compared.

→ As requested, we elaborated on the comparison between early and late gastric cancer patients as follows.

Page 25, line 445; We listed blood metabolites that were significantly altered in CRCA mice (N = 13) compared with control C57BL/6N mice (N = 21), and in advanced gastric cancer patients diagnosed with cachexia (N = 25) compared with early gastric cancer patients without cachexia (N = 14; Table S3). We then compared the mouse and human short lists and restricted them to 15 metabolites that exhibited similar changes in CRCA mice and advanced gastric cancer patients with cachexia (Fig. 8a).

Similarly, the text surrounding malonylation/acetylation after 6 months in patients should be clearer – at current, it is difficult to appreciate that these are serial blood samples from the same patient, with comparisons made based upon weight loss over that 6-month period.

→ We added a description of the analyzed samples to the legend of Fig. 8e. We also

assigned ID numbers to clinical samples to clarify that they were from four patients and not consecutive samples from the same patient.

Page 82, line 1441; Change values were calculated from SEKI mouse datasets (N = 3) with control nude mice (N = 3), CRCA mice (N = 3) with control C57BL/6N mice (N = 3), and gastric cancer cachexia patients (N = 4) with control early gastric cancer patients without cachexia (N = 6).

Page 82, line 1445; each patient's

Reviewer #2 (Remarks to the Author):

The authors have addressed my concerns and have significantly improved their manuscript, in particular in regard to better describing the mouse models that were used in the study. This was a very crucial point.

→ We are pleased that we were able to address the reviewer's concerns.

Since the first revision of the manuscript, a new study has appeared describing NAD⁺ metabolism and niacin in the cachexia context (PMID: 37012289) which should be critically addressed here, since the findings therein are important for the current study.

→ As described in our response to Reviewer 1's comment, we have added this work (PMID: 37012289; Beltrá M. et al., 2023, Nature Communications) and other relevant papers to the reference list.

Page 7, line 118; PMID: 27153497, PMID: 32298240, PMID: 32599075, PMID: 34940589, and PMID: 37012289.

Page 11, line 191; PMID: 9703363, and PMID: 35705545

We thank the reviewer for his/her suggestion to critically address Beltrá M. et al. (PMID: 37012289). Their findings are certainly important; however, they rely heavily on a FOLFOX-treated Colon26 xenograft model, a relatively complicated model that mimics chemotherapy-induced cachexia rather than simple cancer cachexia. Indeed, we have also observed that 5FU, a key drug in the FOLFOX protocol, can decrease muscle NAD, even in wild-type C57BL/6N mice without tumors (Fig S4a; Table S2). 5FU affects C1 metabolism by inhibiting thymidylate synthase, and we have found that hepatic C1 metabolism is closely linked to NAD metabolism (Fig. S4). Therefore, we think that further extensive studies are needed to critically address the work in comparison with our study. Given that their finding of decreased NAD levels in skeletal muscle and liver of cachectic mice is reasonably consistent with our current results, we have limited ourselves to citing this paper at an appropriate location in this manuscript (Page 7, line 118).

Please note that there is a typo in headline of figure 1d (live instead of liver)

→ We thank the reviewer for pointing out the spelling error. We have corrected it.

Reviewer #3 (Remarks to the Author):

The author's addressed my main point and the rebuttal to the other Reviewer's has strengthened the manuscript.

→ We thank the reviewer for this positive assessment.

REVIEWERS' COMMENTS

Reviewer #1 (Remarks to the Author):

Thanks to the authors for their thoughtful consideration of my concerns, as I believe that the revisions have made the manuscript much more accessible to the broad readership of Nature Communications. I have no remaining concerns.

REVIEWER COMMENTS

Reviewer #1 (Remarks to the Author):

Thanks to the authors for their thoughtful consideration of my concerns, as I believe that the revisions have made the manuscript much more accessible to the broad readership of Nature Communications. I have no remaining concerns.

→ We sincerely value the reviewer's helpful advice and constructive criticism, which significantly contributed to strengthening the manuscript.